

# Chemical and spectroscopic characterization of (Artemisinin/Querctin/Zinc) novel mixed ligand complex with assessment of its potent high antiviral activity against SARS-CoV-2 and antioxidant capacity against toxicity induced by acrylamide in male rats

Samy M. El-Megharbel[1,*], Safa H. Qahl[2], Bander Albogami[3] and Reham Z. Hamza[3,*]

[1] Department of Chemistry, College of Sciences, Taif University, Taif, Saudi Arabia
[2] Department of Biology, College of Science, University of Jeddah, Jeddah, Saudi Arabia
[3] Biology Department, College of Sciences, Taif University, Taif, Saudi Arabia
[*] These authors contributed equally to this work.

Corresponding author
Reham Z. Hamza,
dr_reham_z@yahoo.com,
Reham.z@tu.edu.sa

## ABSTRACT

A novel Artemisinin/Quercetin/Zinc (Art/Q/Zn) mixed ligand complex was synthesized, tested for its antiviral activity against coronavirus (SARS-CoV-2), and investigated for its effect against toxicity and oxidative stress induced by acrylamide (Acy), which develops upon cooking starchy foods at high temperatures. The synthesized complex was chemically characterized by performing elemental analysis, conductance measurements, FT-IR, UV, magnetic measurements, and XRD. The morphological surface of the complex Art/Q/Zn was investigated using scanning and transmission electron microscopy (SEM and TEM) and energy dispersive X-ray analysis (XRD). The *in vitro* antiviral activity of the complex Art/Q/Zn against SARS-CoV-2 and its *in vivo* activity against Acy-induced toxicity in hepatic and pulmonary tissues were analyzed. An experimental model was used to evaluate the beneficial effects of the novel Art/Q/Zn novel complex on lung and liver toxicities of Acy. Forty male rats were randomly divided into four groups: control, Acy (500 mg/Kg), Art/Q/Zn (30 mg/kg), and a combination of Acy and Art/Q/Zn. The complex was orally administered for 30 days. Hepatic function and inflammation marker (CRP), tumor necrosis factor, interleukin-6 (IL-6), antioxidant enzyme (CAT, SOD, and GPx), marker of oxidative stress (MDA), and blood pressure levels were investigated. Histological and ultrastructure alterations and caspase-3 variations (immunological marker) were also investigated. FT-IR spectra revealed that Zn (II) is able to chelate through C=O and C-OH (Ring II) which are the carbonyl oxygen atoms of the quercetin ligand and carbonyl oxygen atom C=O of the Art ligand, forming Art/Q/Zn complex with the chemical formula $[Zn(Q)(Art)(Cl)(H_2O)_2] \cdot 3H_2O$. The novel complex exhibited a potent anti-SARS-CoV-2 activity even at a low concentration ($IC_{50} = 10.14 \, \mu g/ml$) and was not cytotoxic to the cellular host ($CC_{50} = 208.5 \, \mu g/ml$). Art/Q/Zn may inhibit the viral replication and binding to the angiotensin-converting enzyme-2 (ACE2) receptor and the main
protease inhibitor (M$^{Pro}$), thereby inhibiting the activity of SARS-CoV-2 and this proved by the molecular dynamics simulation. It alleviated Acy hepatic and pulmonary toxicity by improving all biochemical markers. Therefore, it can be concluded that the novel formula Art/Q/Zn complex is an effective antioxidant agent against the oxidative stress series, and it has high inhibitory effect against SARS-CoV-2.

# INTRODUCTION

Among viruses belonging to the coronavirus family, severe cases globally are usually associated with SARS-CoV-2, the causative agent of the coronavirus disease-2019 (COVID-19) pandemic. Despite numerous preventive and curative measures taken during the recent pandemic, novel infectious variants of SARS-CoV-2 have been recently identified (*Mohapatra et al., 2022c*; *Mohapatra et al., 2022b*).

Reasons which may be attributed to the facilitation of it's transmission include overcrowding, defying social distancing, and an elevated immunodeficient population. Such situations forward the viral replication and increased mutation risk leading to the emergence of novel variants (*Mohapatra et al., 2022a*; *Dhawan, Priyanka & Choudhary, 2022*).

Few drugs can alleviate the symptoms of COVID-19. Drugs used for treating viral infections should be safe, effective, and economical. Coordination compounds with primary transition metal ions have gained high importance for their crucial key roles in the field of therapeutics, material sciences and biological sciences because of their availability with low cost (*Mohammad et al., 2022*). Zinc, a vital bio-metal in nature, has excellent biological and catalytic properties making it one of the best candidates in the field of combating against SARS-CoV-2 (*Mohammad et al., 2021*).

COVID-19 was caused by SARS-CoV-2, a member of the Coronavirus family, which contains four main spikes (*Mohammad et al., 2022*). Spikes play a vital role in the attachment, fusion, and entry of SARS-CoV-2 virus (*Mohammad et al., 2021*). This viral family has threatened human beings and has been announced as a pandemic emergency by WHO (*Rajan et al., 2021*). SARS-CoV-2 contains four main proteins: spike (S), membrane (M), envelope (E), and nucleocapsid (N). Protein S plays a crucial role in the attachment, fusion, and entry of SARS-CoV-2 virus. The S1 domains of the SARS-CoV-2 virus contain receptor-binding domains (RBD) that directly bind to the host cellular receptors (*Xinyu et al., 2023*).

Based on this concept, therapies that directly target SARS-CoV-2 are anticipated to have the greatest effect early in the course of the disease, while immunosuppressive/anti-inflammatory therapies are likely to be more beneficial in the later stages of COVID-19 (*Xinyu et al., 2023*). The clinical spectrum of SARS-CoV-2 infection includes asymptomatic or pre-symptomatic infection and mild, moderate, severe, and critical

## SARS-CoV-2 Entry through Host ACE2

**Figure 1** SARS-CoVs recognizes angiotensin-converting enzyme-2 (ACE2).

illness. Despite the urgent global need and World Health Organization recommendations, a therapy of COVID-19 is currently limited and includes only few drugs: remdesivir, dexamethasone, tocilizumab, ritonavir-boosted nirmatrelvir (Paxlovid), sotrovimab and molnupiravir (*Al-Shemary et al., 2023*). This includes patients who do not require hospitalization or supplemental oxygen and those who have been discharged from an emergency department (*NIH, 2022*).

SARS-CoV-2 and its variants of great concern that infects the respiratory tract, kidney, liver, heart and nervous system and may lead to series of multiple organ failure (*Mohapatra et al., 2022d*). However, the recently emerged Omicron variant reduces the protective effect of vaccination and showed immune-evasive property and it is currently the dominant strain of SARS-CoV-2 around the globe (*Ranjan et al., 2022*).

Additionally, the surface of SARS-CoV-2 contains S1 domains, which comprise RBD and S2 that encompass the fusion peptide. SARS-CoV-2 can enter the host cell through an interaction between the RBD and SARS-CoV-2, which forms a SARS-S-RBD complex with the cellular surface of the host through its receptor angiotensin-converting enzyme-2 (ACE2), as shown in Fig. 1 (*Babcock et al., 2004*; *Wong et al., 2004*).

The RBD of SARS-CoV S protein is located in the S1 subunit and facilitates the binding of SARS-CoV-2 to its host cellular receptors (*Anand et al., 2003*; *Perlman & Netland, 2009*). The interaction of SARS-CoV-2 and ACE2 is crucial for its pathogenesis. The inhibition of this binding can prevent the development of infection. Traditional medicinal plants produce compounds that are considered active therapeutic agents against several pathogens (*Dimitrov, 2003*).

The main scientific efforts towards the discovery of new drugs to combat against SARS-CoV-2, sometimes by using computer-based methodologies that focused on targeting the main active receptors of SARS-CoV-2 such as: $M^{pro}$ and ACE2 (*Mohnad et al., 2021*).

Phytochemicals were excessively studied to find potential inhibitors of $M^{pro}$ *via* a virtual structure-based drug approach (*Joshi et al., 2020*) reporting molecular docking (MD) simulations and toxicity profiles of these derivatives (*Şevki et al., 2021*).

SARS-CoV-2 infection resulted in a lot of millions of confirmed infected cases. COVID-19 pandemic affected a lot of organs of the body through induction of oxidative damage. So, an effective antiviral agent is urgently needed to combat against COVID-19 pandemic (*Hamza et al., 2021*).

Liver diseases and their complications are a major health concern and are associated with a high mortality rate worldwide. Owing to the lack of effective treatment agents, SARS-CoV-2 infections can cause severe oxidative damage to the liver (*Xiao et al., 2003*).

There are several types of liver diseases: viral hepatitis, cholestasis, liver fibrosis, liver cirrhosis, and primary hepatic tumors, including hepatocellular carcinoma (HCC) (*Iyengar, 1985*; *Ye & Jianrong, 2021*; *Hardy & Mann, 2016*; *Pimpin et al., 2018*; *Wang et al., 2014*). The liver is a primary detoxification organ that protects against the toxicity of any xenobiotics. However, it is also vulnerable to COVID-19 and environmental xenobiotics (*Grant, 1991*).

Several patients have succumbed to hepatic diseases, and hepatitis B virus-induced liver cirrhosis primarily leads to liver-related deaths (*Sarin et al., 2020*). The progression of chronic liver disease to decompensated hepatic cirrhosis drastically limits the chances of effective treatment. Therefore, it is crucial to explore alternative effective drugs, especially considering the progressive prevalence of hepatic dysfunction diseases.

*Artemisia annua* L. can effectively treat malaria, and inflammatory diseases, possess antimicrobial activities, and has a good safety record (*Iftekhar et al., 2022*).

Artemisinin (Art) (*Artemisia annua* L.) is a derivative and it belongs to a family of drugs approved for the treatment of malaria with known clinical efficacy. Additionally, Art displays anti-viral and anti-cancer effects. Recently, much more attention has been paid to the miracle key role of artemisinin in treatment of hepatic diseases. Several studies suggested that Art can protect the hepatic tissues from different hepatic dysfunction as proliferation and metastasis (*Iftekhar et al., 2022*).

*Artemisia annua* L. has a beneficial and safety record in the treatment of malaria, antimicrobial activities, hyperlipidemia and inflammatory diseases (*Iftekhar et al., 2022*).

Artemisinin (Art), which is an active ingredient, has an excellent low-toxicity safety profile and is relatively inexpensive. Additionally, these drugs have potential and effective properties. Art can be used either alone or combined with other drugs to elevate

its therapeutic effectiveness and share essentially in the retardation of cases of drug resistance (*Iftekhar et al., 2022*). Thus, the world needs the discovery of new antiviral drugs and an effective strategy to treat emerging diseases that could be trusted safely.

Acrylamide (Acy) is found in starchy foods that had been heated over long periods over 120 °C (*Hamza, Al-Motaani & Malik, 2019*). Acy is formed in the Maillard reaction, non-enzymatic glycation of proteins (*Hamza, Al-Motaani & Malik, 2019*). This reaction occurs between the amine group residue ($-NH_2$) of proteins and the carbonyl group ($C=O$) from carbohydrates when food is heated above 120 °C.

Acy is not a food additive, but it is a by-product of the cooking processes and could be produced when the food is cooked at very high temperatures. Acy was classified as a Group 2A carcinogen by the international Agency for Research on carcinogens, In previous experimental rat models, Acy led to hepatotoxicity (*Hamza, AL-thubaiti & Omar, 2019*), lung toxicity (*Kerim et al., 2022*; *Zeynep et al., 2022*), and testicular damage (*Erdemli et al., 2019*).

Hepatic disorders have serious side effects. So, it is necessary to seek for new safe medicines for the liver diseases especially, those originating from natural resources (*Hamza, Al-Motaani & Malik, 2019*).

Quercetin (Q) is a natural flavonoid widely found in plants and confers several benefits, such as antioxidant and anti-inflammatory activities. Additionally, the administration of Q minimizes oxidative damage. Q supplementation tremendously helps in restoring of the normal levels of blood glucose and lowering serum cholesterol levels. Additionally, Q enhances antioxidant capacities and prevents oxidative damage (*Hamza, Al-Motaani & Malik, 2019*).

Q protects and alleviates inflammatory storms (*Zarezade et al., 2018*; *Refat et al., 2021*; *Vessal, Hemmati & Vasei, 2003*), particularly those caused by viral infections. Nutrients promote the generation of an immune response. The role of micronutrient metals, such as zinc (Zn), in treating of many inflammatory diseases has been extensively evaluated (*Refat et al., 2021*; *Maret, 2017*).

There is a strong correlation between Zn levels and organ dysfunction owing to the association of several diseases with Zn transporter polymorphisms *via* the ZnT8 gene (*Maret, 2017*). Zinc plays a key role in treating of the pulmonary diseases (*Lassi, Moin & Bhutta, 2016*). Zn supplementation inhibits pneumonia, and it can increase resistance against infections (*Lassi, Moin & Bhutta, 2016*).

The previous study revealed that synthesized ZnO–NPs have inhibitory activity against SARS-CoV-2 when used as a nanospray against SARS-CoV-2 infection; however, it had some cytotoxic activity (*El-Megharbel et al., 2021*). Confirming our concepts, the recent study also demonstrated that (Artemisinin/Zn) novel complex shows a moderate inhibitory and antiviral effects against SARS-CoV-2 with a value of IC50 equal to 66.79 µg/ml. So, novel Art-complexes may strengthen the inhibitory and antiviral effects the SARS-CoV-2 with very low cytotoxicity (*Al-Salmi, El-Megharbel & Hamza, 2023*).

The current study was conducted to characterize the spectroscopic and chemical structure of Art/Q/Zn synthesized novel complex, evaluate its cytotoxic, inhibitory and antiviral activities against the SARS-VoV-2, assess its antioxidant capacities, analyze its

**Figure 2  Chemical structure of (A) quercetin and (B) artemisinin.**

ability to alleviate hepatotoxicity and pulmonary toxicity induced by Acy in male rats, and propose this complex as a promising novel formula that could inhibit SARS-CoV-2 with high levels of safety.

# MATERIALS AND METHODS

## Ethical approval

The inhibitory effect of Art/Q/Zn novel synthesized complex against the SARS-CoV-2 was tested at the Centre of Scientific Excellence for Influenza Viruses, National Research Centre, Dokki, Egypt using the SARS-CoV-2 isolate hCoV-19/Egypt/NRC-03/2020 (EMBL-EBI, accession number SAMN14814607). The biological activity of the novel complex Art/Q/Zn was tested in male rats. The experiment was approved by the ethical approval committee of Zagazig University (approval number ZU-IACUC/2/F/61/2022). This study was also approved for evaluation of Art on lung functions by the Taif University ethical committee, accredited by the national committee for Bioethics (approval No. HAO-02-T-105). The SARS-CoV-2 strain sample used in the research has ethical approval and the accession number: (hCoV-19/Egypt/NRC-03/2020 (Accession Number on GSAID: EPI_ISL_430820). The bio-sample's record history can be accessed through the link: https://www.ebi.ac.uk/biosamples/samples/SAMN14814607. The available phylogenetic analysis is accessible at: http://purl.obolibrary.org/obo/NCBITaxon_2697049.

## Chemicals and instrumental analyses

Quercetin (Q, CAS Number: 117-39-5) (Fig. 2A), artemisinin (Art, CAS Number: 63968-64-9, Fig. 2B), zinc chloride ($ZnCl_2$, CAS Number: 7646-85-7), and acrylamide (Acy, CAS Number: 79-06-1) were obtained from Sigma-Aldrich (St. Louis, MO, USA) and then used with no more purification. The analytical instruments employed are mentioned in Table 1.

**Table 1  Tool and instrumental analyses.**

| Analysis instrumentation | Models |
|---|---|
| Elemental analysis | PerkinElmer CHNS 2400 Elemental analyzer |
| Conductance | Jenway 4010 conductivity meter |
| FTIR spectra | Bruker FTIR Spectrophotometer |
| Electronic spectra | ATi Unicam UV2 Cary 60 UV/Vis. Spectrophotometer |
| TEM | JEOL 100s microscopy |
| Magnetic measurements | Sherwood Scientific Magnetic Balance using Gouy method |
| SEM | Quanta 250 FEG (SEM) Scanning Electron Microscope |
| X-ray diffraction patterns | X 'Pert PRO PANanalytical X-ray powder diffraction, target copper with secondary monochromate. |

## Synthesis of Artemisinin/Quercetin/Zinc novel mixed ligand complex

The Art/Q/Zn complex was formed and produced by mixing an ethanolic solutions of zinc chloride ($ZnCl_2$) (0.001 mole, 0.137 g) with Q (0.001 mole, 0.302 g) and Art (0.001 mole, 0.283 g). Next, this mixture was completely dissolved in 25 ml of ethanol. The pH was adjusted to 8.0–9.0 using 10% ammonia solution. The solution was refluxed for 5 h using a refluxing system. Next, the solution was filtered using a semi-thin filter (25 µm) and washed with ethanol (96%) and purified distilled water. Finally, the resulting solid mixed ligand complex was dried in an oven at 80 °C and then kept in a desiccator (Fig. 3).

## Cytotoxicity concentration assay

To assess the maximal ($CC_{50}$) half cytotoxic concentration of the newly synthesized complex of Art/Q/Zn, a stock solution of Art/Q/Zn was produced by using a 10% mixture of Dimethyl sulfoxide (DMSO), diluted with DMEM and di-distilled water. The cytotoxic activity of Art/Q/Zn was tested on VERO-E6 cells by using the MTT assay as previously described in our previous study (*Al-Salmi, El-Megharbel & Hamza, 2023*). Briefly, VERO-E6 cells were seeded at 100 µl per each well and were all incubated at 37 °C in a $CO_2$ environment for approximately 24 h. Following this, varying concentrations of Art/Q/Zn were administered to VERO-E6 cells in triplicates. VERO-E6 cells were all treated with Art/Q/Zn novel complex at different concentrations and tested in 3 times (triplicates). Subsequently, the obtained supernatants were suddenly discarded, and the cellular monolayers underwent three washes using MTT solution (5 mg/mL) of Art/Q/Zn solution in combination with sterile phosphate buffer saline. The cells were then incubated at 37 °C for 4 h. The formazan crystals were dissolved in 200 µl of isopropanol and HCl mixture. The absorbance of the formazan crystals was measured within the wavelength range of 540 to 620 nm using a multi-well plate reader. The calculation of cytotoxicity percentage was performed according to the equation:

% Cytotoxicity = (absorbance of cells without Art/Q/Zn (Control) − absorbance of VERO-E6 cells with Art/Q/Zn) ×100/absorbance of VERO-E6 cells without Art/Q/Zn.

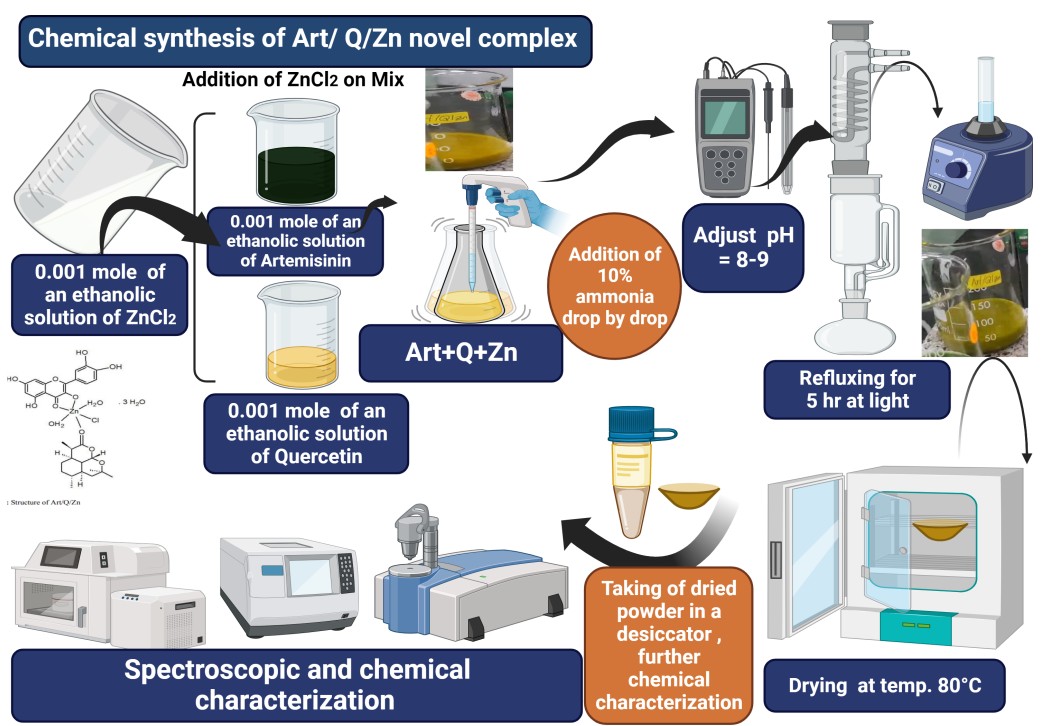

**Figure 3** Art/Q/Zn novel mixed ligand complex.

## Determination of ($IC_{50}$) inhibitory concentration 50%

In tissue cultures' healthy plates, the Vero-E6 cells were all distributed and incubated (Temp. 37 °C) under $CO_2$ conditions overnight. Three times with phosphate buffer and then exposed to the virus adsorption (hCoV-19/Egypt/NRC-03/2020 (EPI-ISL-430820)) for 1 hr at the room temperature (RT). At 37 °C for only 1 hr, the monolayers were overlaid with DMEM media containing concentrations of Art/Q/Zn. After incubation at 37 °C for 72 h, VERO-E6 cells were treated with paraformaldehyde that was freshly prepared as (4% paraformaldehyde/PBS), just warming PBS (Phosphate buffer saline) then, with continuous vigorous stirring, slowly add paraformaldehyde, then gradually add a few drops of 0.1 M NaOH based on (*O'Neil, 2006*) that confirmed that paraformaldehyde dissolved in fixed alkali hydroxide solution, then filter with filter paper till obtain solution with pH 7.9 for 20 min at room temperature (RT) and stained with 0.1% crystal violet stain in distilled water for about 15 min. Then, we added methanol for each well to dissolve the crystal violet dye, and the color optical intensity of Art/Q/Zn was then measured at absorbance of 570 nm by using multi-plate readers. The $IC_{50}$ of Art/Q/Zn novel complex is needed to decrease the "SARS-CoV-2" cytopathic effect by 50% percentage ratio, which is relative to control SRAS-CoV-2 (*El-Megharbel et al., 2021*).

## Treatment of animals and animal ethics approval

The animals were treated using a procedure approved by the ethical approval committee of both Zagazig and Taif Universities (the ethical approval numbers ZU-IACUC/2/F/61/2022

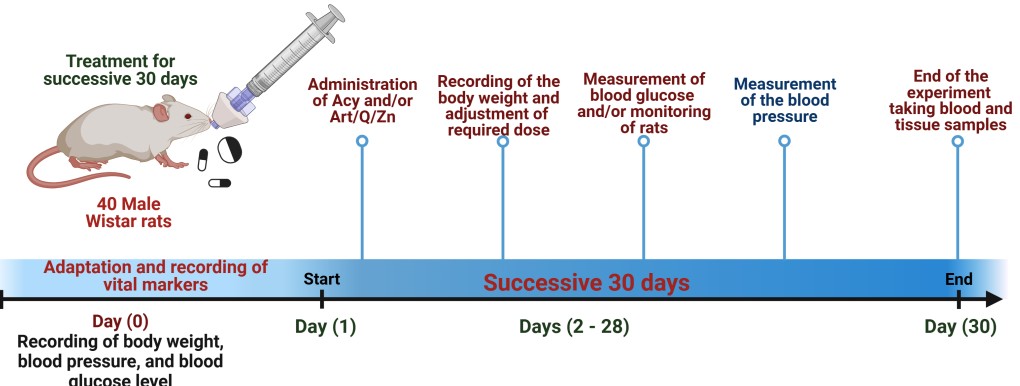

**Figure 4  Timeline for the *in vivo* experimental design.** Experimental timeline for (Acy) and novel complex (Art/Q/Zn) treatment.

and HAO-02-T-105, respectively). Male rats weighing 150–180 g were obtained from the animal house of the Faculty of Pharmacy, Zagazig University. Healthy pathogen-free animals aged 6 weeks were then immediately kept at the animal house of the Zoology Department (Physiology division), Faculty of Science. The animals were housed under standard laboratory conditions (27 °C and a normal daylight cycle). Food and water were provided *ad libitum*.

The experiment was conducted in 40 male rats who were randomly divided into four groups. The rats were administered the treatments *via* the oral route using an oral gastric tube for 30 days. The treatment administered in the four groups is mentioned as follows:

Group I: control group: animals received normal physiological saline (0.9% NaCl).

Group II: animals were administered Acy (500 mg kg$^{-1}$) in saline solution (*Hamza, Al-Motaani & Malik, 2019*).

Group III: animals were administered Art/Q/Zn (30 mg Kg$^{-1}$) that was dissolved in normal physiological saline (*Refat et al., 2021*).

Group VI: animals were orally administered Acy and the Art/Q/Zn complex after 30 min, as described previously, (Figs. 4 and 5) for 30 successive days.

## Samples collection

Blood samples were centrifuged at 5000 r.p.m for about 5 min to obtain the serum. The serum was used for further analysis, and it was persevered at −20 °C. The experimental male rats were suddenly decapitated after light anesthesia with xylene/ketamine (I.P), and both the liver and lung tissues were removed, 1$^{st}$ part of liver and lung tissues was immediately used for the histological, ultra- structural and immune testing, Meanwhile the 2$^{nd}$ part of liver tissues was weighed, and homogenized and used immediately for the evaluation of antioxidant enzyme capacities (Fig. 1). The supernatants of hepatic tissue homogenates were obtained after centrifugation at 3,000 g for 1/4 hr at 4 °C, then the supernatants were immediately collected and preserved at −20 °C for further analysis.

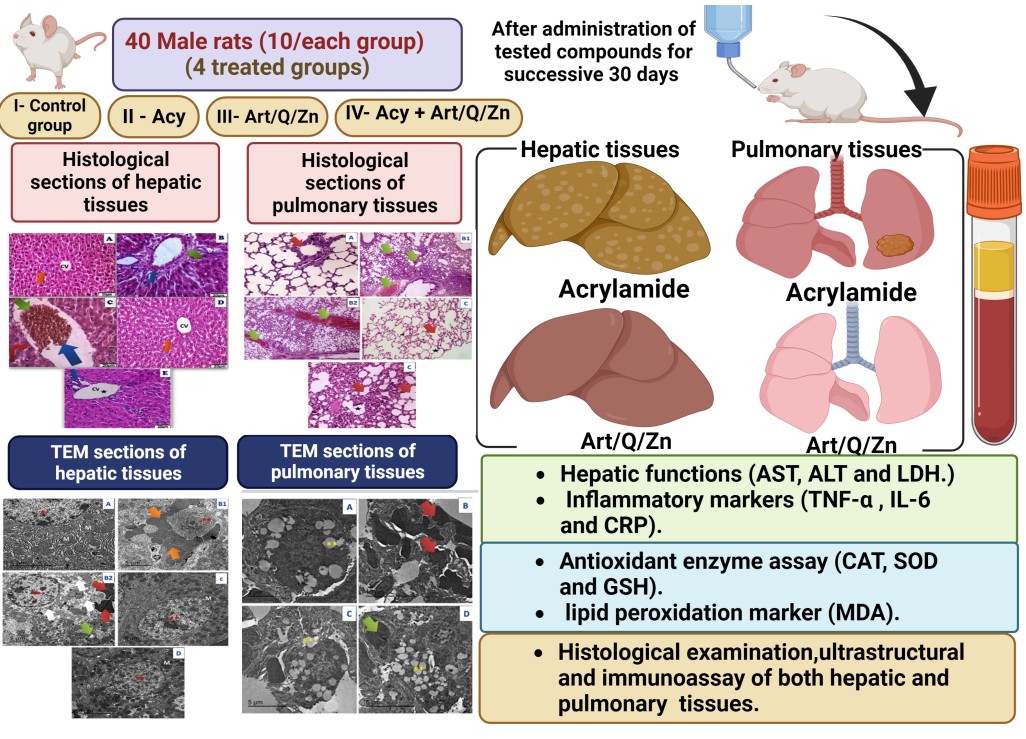

**Figure 5** Schematic representation of the experimental design.

## Biochemical investigation

### Measurement of hepatic functions and inflammation markers

After treatment for successive 30 days, some biochemical markers in serum were assessed as the activity of alanine aminotransferase (ALT), aspartate aminotransferase (AST) by using available commercial kits from (Spinreact Co, Spain) according to the instructions. Serum LDH activity was assessed by using LDH kit (GmbH Schigraben, Hannover, Germany).

The ELISA technique was conducted (Ebio-Science) by following the instructions to determine tumor necrosis factor-alpha (TNF-$\alpha$), interleukin-6 (IL-6) and determination of C-reactive protein (CRP).

### Assessment of oxidative stress markers

A small piece (0.25 g) of the hepatic tissues were homogenized with cold buffer and it was centrifuged to get the supernatant that was further used for performing antioxidant assays. According to *Ohkawa, Ohishi & Yagi (1979)*, malondialdehyde (MDA) was measured.

Superoxide dismutase enzyme activity (SOD) was assessed according to *Sun, Oberley & Li (1988)*, CAT activity was estimated according to Aebi, and the breakdown rate of hydrogen peroxide was at 240 nm (Spectrophotometer SP-2200, BioSpectra, Bangor, PA) (*Aebi, 1984*), it was expressed in (U/g). Glutathione peroxidase (GPx) activity was evaluated according to the manufacturer instructions (*Paglia & Valentive, 1967*).

## Histological and immunohistochemically assessment of both liver and lung tissues

Fixed samples were processed by using hematoxylin/eosin staining of liver and lung tissue sections. Photomicrographs of the tissues were captured by light microscope. The excessed liver and lung slices (four mm thickness) were blocked with within 0.1% mix of water and methanol for about 1/4 hr to study apoptosis-related proteins. After the blocking process, both tissue sections were treated at 4 °C overnight with polyclonal caspase-3 antibody. The color intensity of caspase-3 in the immunohistochemical sections was used to classify the intensity as follows: (+) means (weak immunoreactivity), (++) means (moderate immunoreactivity), (+++) means (high and strong immunoreactivity), and (++++) means (very strong immunoreactivity).

## Measurement of blood pressure

Systolic, diastolic and mean arterial blood pressures (BP) were measured using noninvasive BP measurement system (NIBP 250, Serial No: 21202-108, BIOPAC system, Inc., Goleta, CA, USA) (*Abubakar, Ukwuani & Mande, 2015*).

## Molecular docking, physicochemical, and pharmacokinetics studies

In the current study, we analyzed the pharmacophoric features of SARS-CoV ACE2 receptor co-crystallized inhibitor novel complex Art/Q/Zn to synthesize novel compounds using the approach of ligand-based design (*Daina, Michielin & Zoete, 2019*). We synthesized a novel Art/Q/Zn formula that can bind with ACE2 and M$^{pro}$ receptors that bind SARS-CoV-2.

Generalized molecular docking simulations of complex. We used Swiss docking tool and based on the previous study of *Abo Elmaaty et al. (2022)*.

## Statistical and data analysis

The data were expressed as mean $\pm$ SE., a one-way analysis of variance was employed for comparing several groups, using post-hoc test. Statistical significance at $P \leq 0.05$ (*Dean, Sullivan & Soe, 2013*). We assumed that CAT in hepatic tissue homogenates in Acy group *versus* Acy+ Art/Q/Zn group are $2.49 \pm 0.86$ *versus* $4.21 \pm 0.95$ (U/g). At power 80% and confidence level 95%, sample size is 40 (10 in every group). This sample was calculated by OPEN EPI software package (*Dean, Sullivan & Soe, 2013*).

# RESULTS

## Physical measurements data

Zinc chloride salt reacted with the mixed ligands of Artemisinin and quercetin according to this equation: $ZnCl_2$ + Artemisinin (Art) + quercetin (Q) = $[Zn(Q)(Art)(Cl)(H_2O)_2]\cdot 3H_2O$.

The proposed structure for zinc mixed ligand complex $[Zn(Q)(Art)(Cl)(H_2O)_2]\cdot 3H_2O$ (molecular formula: $(C_{30}H_{41}O_{19}ClZn)$) was in (Fig. 6) and elemental analysis values: (%C = 44.72, %H = 5.09, %Cl = 4.40) which shows that the molar ratio of Zn:Q:Art is 1:1:1. The novel mixed ligand Art/Q/Zn complex was stable in the air and was soluble in

**Figure 6** Structure of Art/Q/Zn novel complex.

DMSO. The value of molar conductivity ($\Lambda_m$) was 23 ohm$^{-1}$·cm$^2$·mol$^{-1}$, supporting the non-electrolytic nature of the mixed ligand complex (*El-Megharbel & Hamza, 2022*).

**Infrared spectra**

Infrared for quercetin (Q), artemisinin (Art), and their zinc complexity are shown in Fig. 7, the assignments for prominent vibrational bands:

For free ligand (quercetin), broad, strong bands appear at 3,563 and 3,388 cm$^{-1}$, referring to hydroxyl (OH) stretching vibration for polyphenolic groups. For Q with Zn(II), a robust and broadband appeared at 3,372 cm$^{-1}$ referring to molecules of water, which is convenient with the suggested structure (Fig. 6) of mixed ligand complex $[Zn(Q)(Art)(Cl)(H_2O)_2]\cdot 3H_2O$. Also, for free (Q), the $\nu(C4{=}O)$ stretching vibration of the carbonyl group appeared at 1,679 cm$^{-1}$, while for $[Zn(Q)(Art)(Cl)(H_2O)_2]\cdot 3H_2O$ complexity, shifting occurred for this band in the range 1,613–1,660 cm$^{-1}$. It confirmed

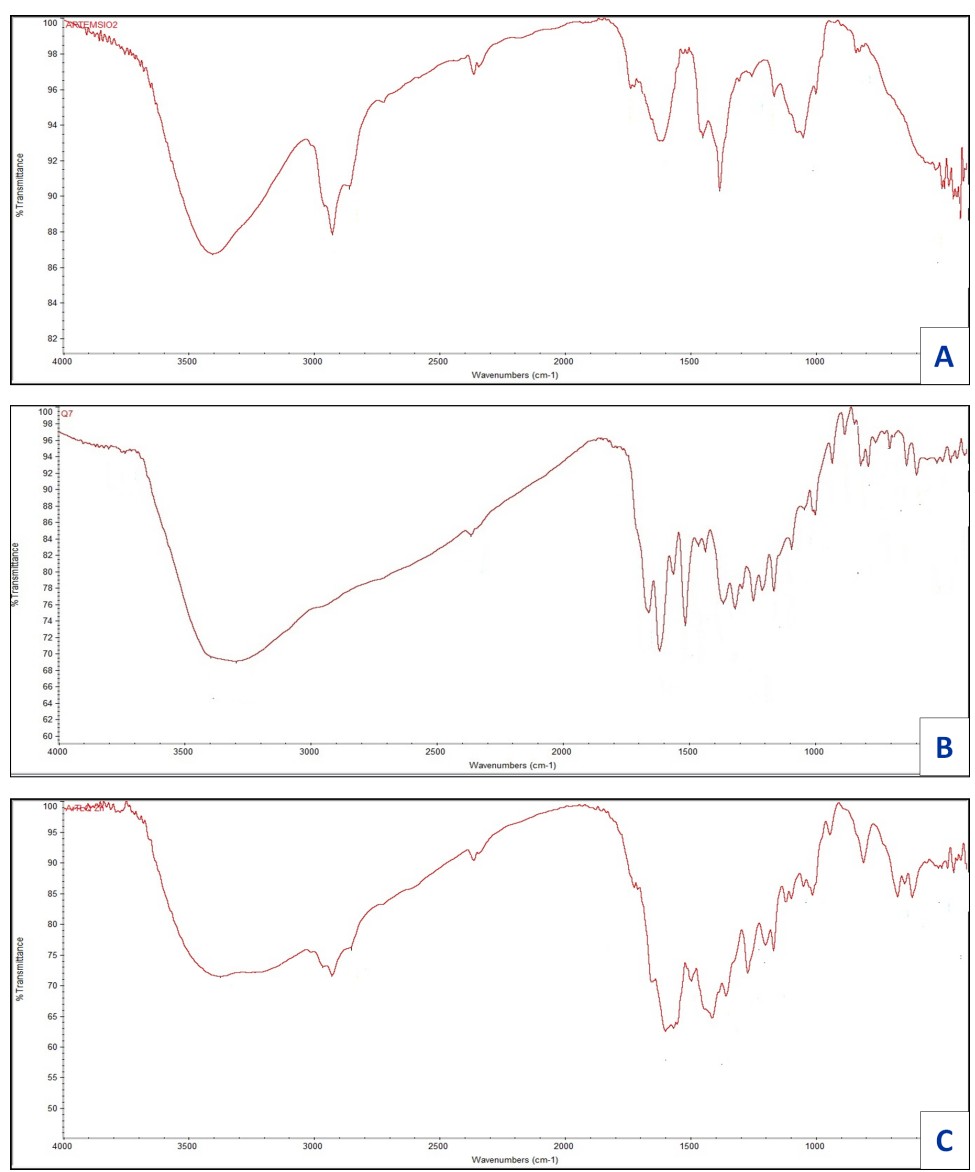

**Figure 7** FT-IR of (A) Art , (B) Q and (C) Art/Q/Zn.

that Zn (II) can be chelated through C(4)=O and C(3)−OH which are the carbonyl oxygen atom of of the (Q) ligand (*Nakamoto, 1986*). The stretching vibration band for ether group $\nu$(COC) of in the ring II for (Q) appeared at 1,269 cm$^{-1}$, where no shift occurred for this band after chelation. Also, ether group $\nu$(COC) in the ring of Art appeared at 1,260 cm$^{-1}$. No change occurred for this band after chelation, which confirmed that this band is not involved in coordination. The $\nu$(C−OH) for free (Q) appears at 1,409 cm$^{-1}$, where shifting occurred from 1,361–1,383 cm$^{-1}$ for [Zn(Q)(Art)(Cl)(H$_2$O)$_2$]·3H$_2$O complexity, and this confirms the involvement of C(3)−OH phenolic oxygen group (ring II) in chelation process.

For the Art free ligand, many bands (strong and broad) appeared at 3,752 and 3,692 cm$^{-1}$, referring to $\nu$(O−H) of the formed O—H hydrogen bond between the ring's hydrogen atom and oxygen atom. Also, for free Art, the carbonyl group's $\nu$(C=O) stretching vibration appeared at 1,736, 1,689 cm$^{-1}$. In contrast, for [Zn(Q)(Art)(Cl)(H$_2$O)$_2$]·3H$_2$O complexity, shifting for this band appeared in the range 1,652–1,722 cm$^{-1}$ and confirming that Zn (II) can be bonded *via* carbonyl (O) atom C=O of the Art ligand. Stretching vibration bands for $\nu$(C–H), aliphatic appeared at the range 2,851–2,967 cm$^{-1}$, for free Art ligand and [Zn(Q)(Art)(Cl)(H$_2$O)$_2$]·3H$_2$O complex.

For Art, the vibrational stretching band for ether group $\nu$(COC) has appeared at 1,260 cm$^{-1}$, where no shift occurred after chelation, confirming that this band is not involved in coordination. The [Zn(Q)(Art)(Cl)(H$_2$O)$_2$]·3H$_2$O has new bands appearing within wavenumbers 611–645 cm$^{-1}$, that are referring to stretching vibration for zinc-chloride bond (*Nakamoto, 1970*; *Bellamy, 1975*). The bands appeared at the range 497–511 and 473–449 cm$^{-1}$, referring to $\nu$(Zinc−O), the stretching vibration which confirmed the formation of metal complexity (*Bellamy, 1975*).

## UV–Vis spectra and magnetic measurement

The electronic spectrum of the free ligand quercetin has absorption bands appeared at 290 and 365 nm referring to $\pi \to \pi^{\star}$ and n $\to \pi^{\star}$ transitions. In comparison, Art free ligand has absorption bands appeared at 295 and 354 nm posted to $\pi \to \pi^{\star}$ and n $\to \pi^{\star}$ transitions, respectively (*Lever, 1984*). The UV-Vis spectrum for [Zn(Q)(Art)(Cl)(H$_2$O)$_2$]·3H$_2$O novel complex has two absorption bands at 290 nm and 345 nm due to $\pi \to \pi^{\star}$ and n $\to \pi^{\star}$ transitions respectively (Fig. 8), (Table 2) and regarded as diamagnetic due to formation of the novel complex Art/Q/Zn caused by the conjugated system by complexation that made the bathochromic shift (*Charisiadis et al., 2014*). Other bands have appeared at 415 nm, 440 nm, and 455 nm for zinc-mixed ligands. These bands can be due to the M →L charge transfer transition. The measured magnetic moment value obtained for the novel Zinc mixed ligand complex lies at 3.98 BM, corresponding to the octahedral field.

## $^1$H-NMR spectra

$^1$H-NMR spectrum for free Q ligand shows the following signals: (1H, C5-OH): $\delta$ 12.52, (1H, C7-OH):10.80, (1H, 3-OH): 9.54,(1H, 4′-OH): 9.61, (1H, 3′-OH):9.32, (1H, 2′-H): 7.62, (1H, 6′-H) :7.54,(1H, 5′-H): 6.89,(1H, 8-H):6.37, (1H, 6-H):6.15 (*Sanna et al., 2016*). For mixed ligand complex [Zn(Q)(Art)(Cl)(H$_2$O)$_2$]·3H$_2$O (Fig. 9) the prominent signal bands appeared are: (1H,5-OH): $\delta$ 12.492, (1H, 7-OH):10.806, (1H, 4′-OH):9.67, (1H, 3′-OH):9.383, (1H, 2′-H):7.664, (1H, 6′-H) 7.661, (1H, 5′-H): 7.534,7.521, (1H, 8-H)6.874, (1H, 6-H):6.856, 6.390, 6.387. According to these data, the band appeared at $\delta$ 9.54 for free Q assigned to (1H, 3-OH) disappeared for [Zn(Q)(Art)(Cl)(H$_2$O)$_2$]·3H$_2$O, and other signal bands not shifted, confirming that hydroxyl group at C-3OH is deprotonated and involved in coordination with Zn(II). Also, there is a new signal band appeared at $\delta$ 3.335 ppm for [Zn(Q)(Art)(Cl)(H$_2$O)$_2$]·3H$_2$O,which is assigned to protons of H$_2$O

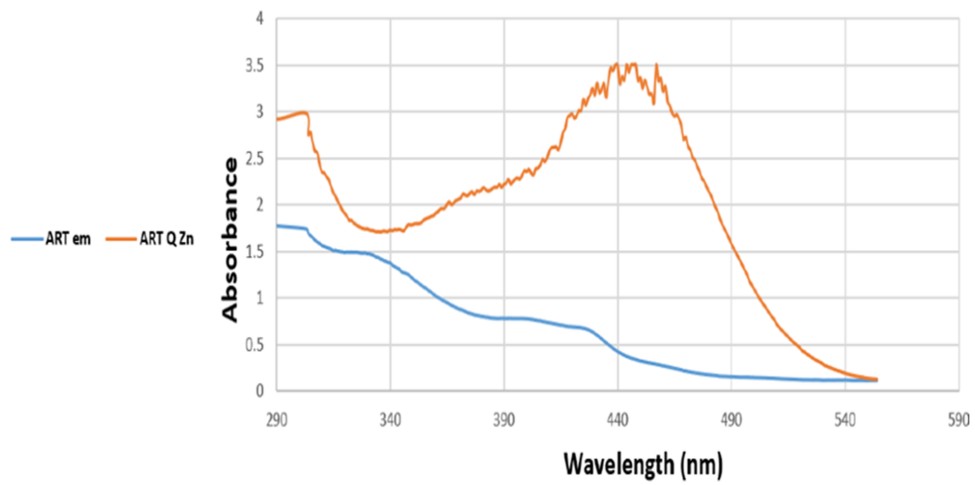

**Figure 8**  UV-Vis spectra of Art and Art/Q/Zn.

**Table 2**  Electronic spectra and magnetic moments of quercetin, artemisinin and their zinc mixed ligand.

| Sample | Electronic bands/ nm | | Magnetic moment | Geometry |
|---|---|---|---|---|
| | $\pi-\pi^{\star}$ | $n-\pi^{\star}$ | | |
| Quercetin | 294 | 365 | - | – |
| Artemisinin | 295 | 354 | - | – |
| $Zn(Q)(Art)(Cl)(H_2O)_2]\cdot 3H_2O$ | 290 | 345 | diamagnetic | octahedral |

coordinated and uncoordinated. Based on the above data, the complexation structure for $[Zn(Q)(Art)(Cl)(H_2O)_2]\cdot 3H_2O$ was confirmed.

## X-ray diffraction analysis

X-ray powder diffraction (XRD) patterns were used to determine the crystallinity of the novel mixed ligand Zinc complex at the value of $(2\theta)$ 4–80° and, also used to examine the nanostructural form of the zinc mixed ligand complex. For X-ray diffractogram of $[Zn(Q)(Art)(Cl)(H_2O)_2]\cdot 3H_2O$, a broad peak at $2\theta = 23°$ appeared, confirming that the zinc mixed ligand complex has an amorphous structure (*Hamza et al., 2022*; *Al-Thubaiti et al., 2022*), as shown in Fig. 10. All our attempts to prepare single crystals failed.

## Scanning electron microscopy SEM and EDX

The technique of SEM determine the microscopic and physical character of Art, (Figs. 11A, 11B); Q, (Figs. 11C, 11D), and $[Zn(Q)(Art)(Cl) (H_2O)_2]\cdot 3H_2O$. (Figs. 11E, 11F). SEM can be taken as an indication for the presence of a single component for $[Zn(Q)(Art)(Cl)(H_2O)_2]\cdot 3H_2O$. The images of $[Zn(Q)(Art)(Cl)(H_2O)_2]\cdot 3H_2O$ clarify nano-feature products with small particle size. The morphological surface for the novel mixed ligand Zinc complex was checked using SEM, showing a small particle with a high

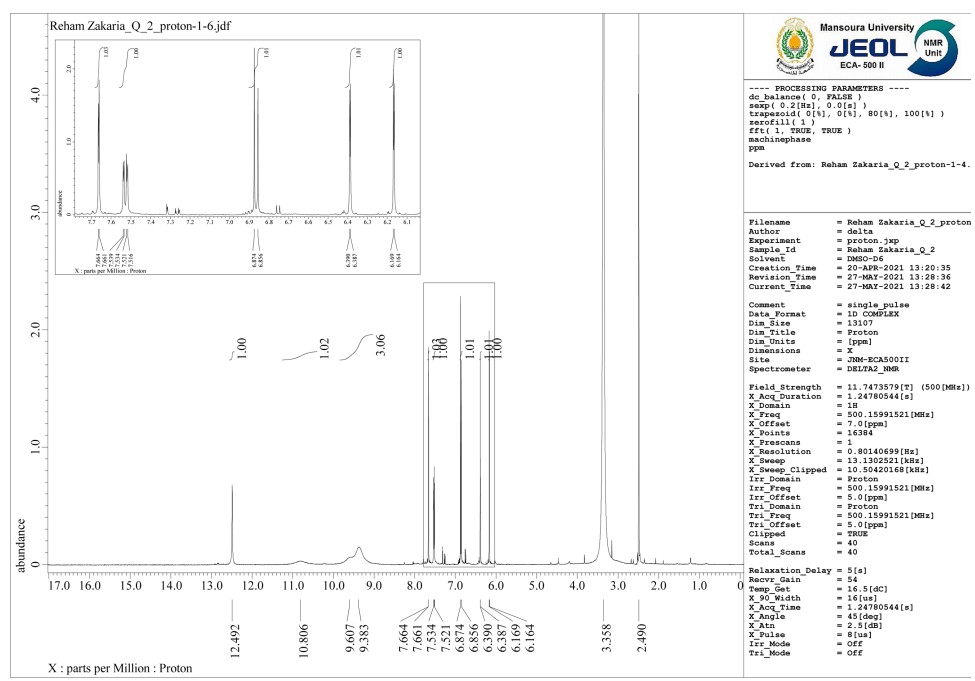

**Figure 9** ¹H-NMR of Art/Q/Zn.

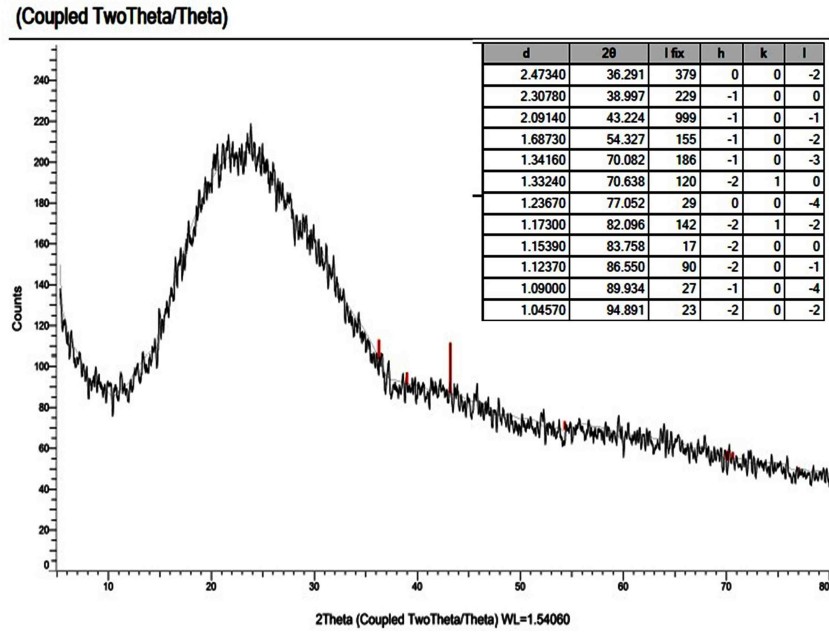

**Figure 10** XRD of Art/Q/Zn mixed ligand.

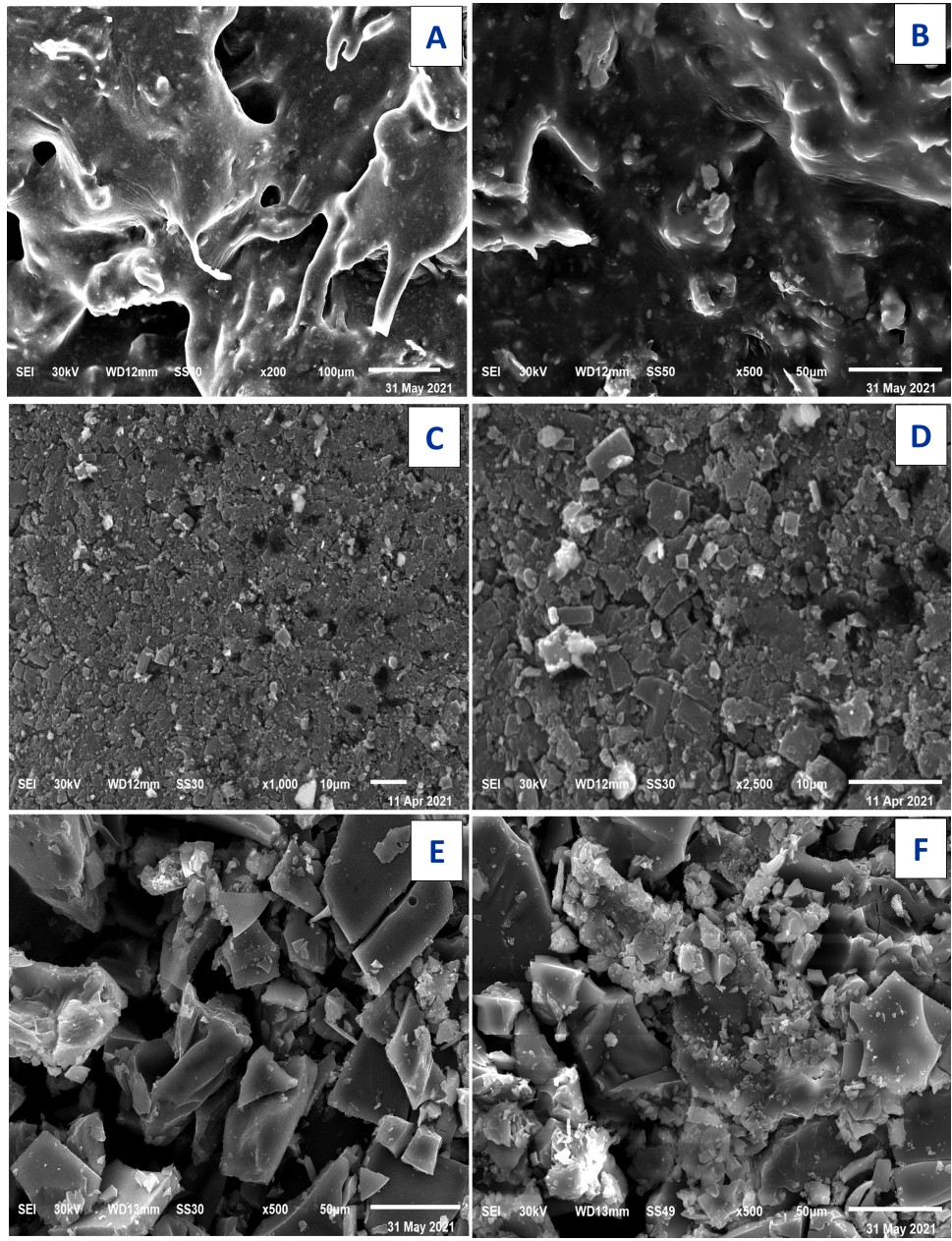

Figure 11    SEM of Art (A,B) , Q (C, D), and Art/Q/Zn (E,F).

more ability to form agglomerates with different shapes. EDX clarified the elemental analysis of the novel complex with percentage (Fig. 12).

## Transmission electron microscopy

TEM images for ligands: free Art, (Figs. 13A, 13B); Q (Figs. 13C, 13D), and synthesized $[Zn(Q)(Art)(Cl)(H_2O)_2]\cdot 3H_2O$ complex Fig. 13 (E,C2). The uniform matrix for $[Zn(Q)(Art)(Cl)(H_2O)_2]\cdot 3H_2O$ complex was cleared in the pictograph and this confirms

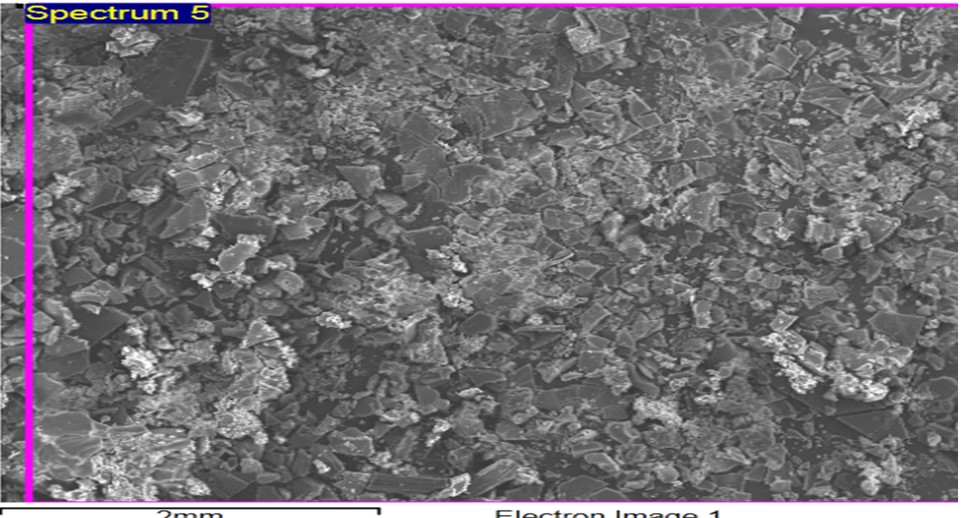

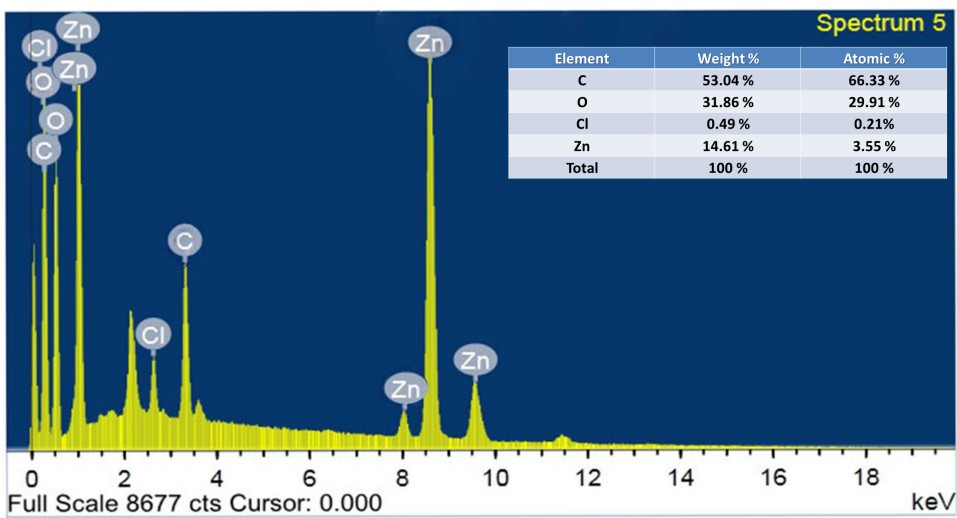

Figure 12 **EDX of Art/Q/Zn mixed ligand complex.**

that $[Zn(Q)(Art)(Cl)(H_2O)_2]\cdot 3H_2O$ complex has a homogeneous phase where spherical black spots like shape is observed for Zn(II) complex with the particle has size ranged of 31.99–48.13 nm.

## Cytotoxicity assay

Art/Q/Zn novel complex showed highly potent inhibitory activity against the SARS-CoV-2 in very low concentration ($IC_{50} = 10.14\ \mu g/ml$). This assay assessed the possible cellular cytotoxicity of Art/Q/Zn ($CC50 = 208.5\ \mu g/ml$) (Fig. 14) and attached an image for crystal violet analysis with silver crystals formed, successive dilution and high anti-viral activity of Art/Q/Zn by appearance of pale purple color as cells that undergo cell death lose their adherence. They are subsequently lost from the population of cells that indicate inhibition of SARS-CoV-2 cell growth.

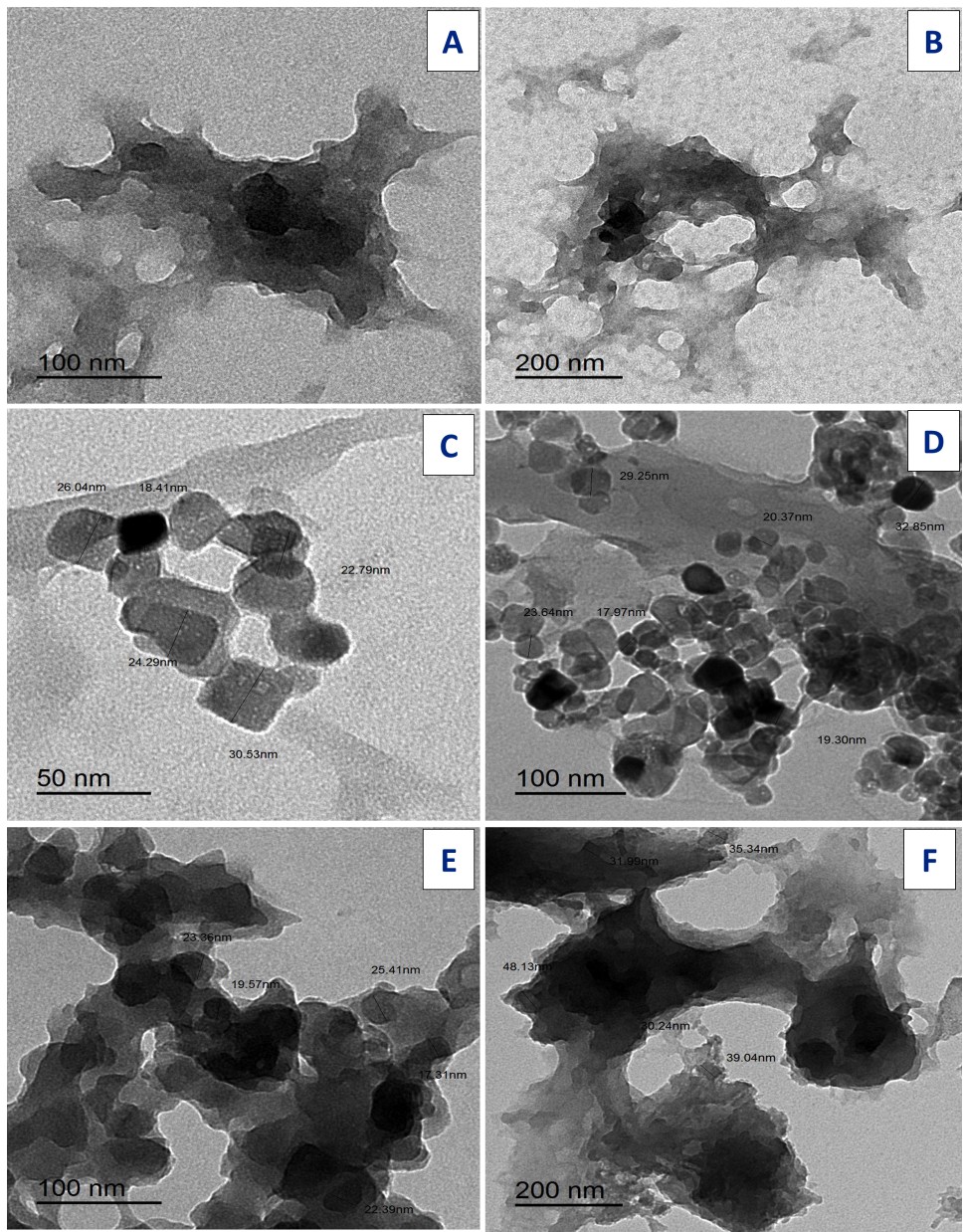

**Figure 13** TEM of Art (A, B) , Q (C, D), and Art/Q/Zn (E, F).

## Biochemical evaluation

AST, ALT, and LDH serum enzyme levels were significantly increased in the Acy group, as shown in Table 3 and Fig. S1. The current findings indicated disruption of the hepatic cellular membranes. Meanwhile, Art/Q/Zn mixed ligand novel complex at a dose of 30 mg/kg was found to be safe for hepatic enzyme activity. Male rats were given Acy and then treated with Art/Q/Zn, which resulted in a decrease in hepatic enzymatic and function activity.

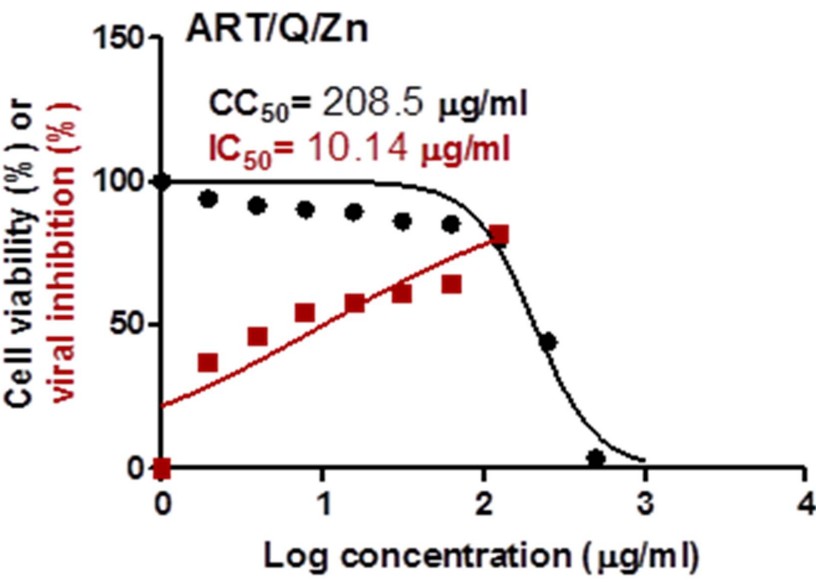

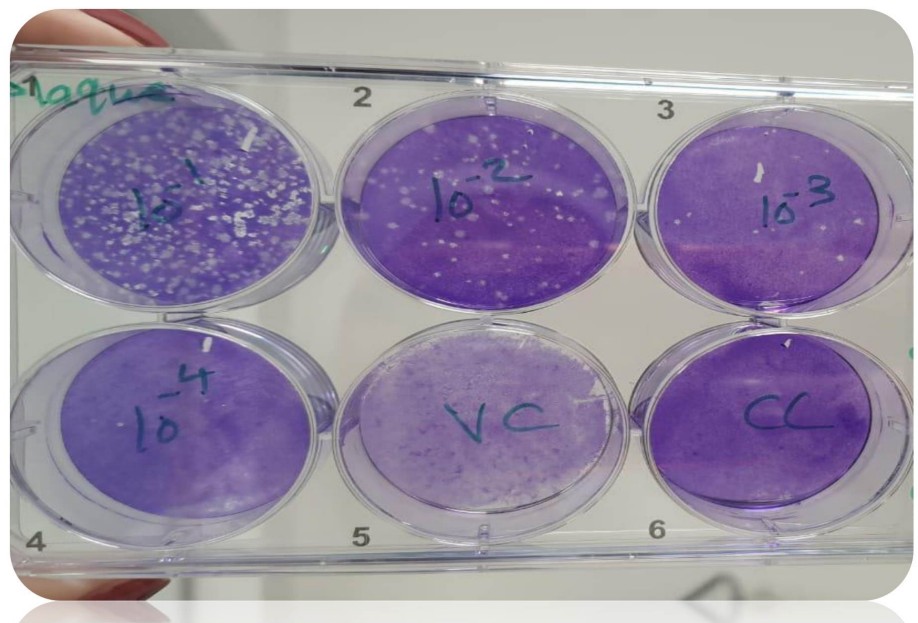

**Figure 14** **Values for the cytotoxicity concentration 50 (CC50) and inhibitory concentration 50 (IC50) (Art/Q/Zn) metal complex.** The figure shows the formula revealing its potent and promising activity against the SARS-CoV-2. The value of IC50 was calculated by the best line drawn between log concentration and viral inhibition % (triplicate/each concentration) to estimate the potent antiviral activity against the SARS-CoV-2 (hCoV-19/Egypt/NRC-03/2020; accession number SAMN14814607) using the Vero E6 cells. (Approved official report from center of scientific excellence for influenza viruses, National research center , Cairo , Dokki, Egypt). Cytotoxicity concentration 50 (CC50): on Vero E6 cells Inhibitory concentration 50 (IC50): Antiviral activity against Severe *Acute Respiratory Syndrome Coronavirus 2 (SARS-CoV-2) (hCoV-19/Egypt/NRC-03/2020).

**Table 3  Assessment of Art/Q/Zn (30 mg/Kg) on hepatic function and enzyme activity of male rats treated with Acy (500 mg/Kg) for successive 30 days.**

| Parameters | Control | Acy (500 mg/Kg) | Art/Q/Zn (30 mg/Kg) | Acy +Art/Q/Zn |
|---|---|---|---|---|
| ALT (U/L) | 13.14 ± 1.09[b] | 184.91 ± 8.42[c] | 13.25 ± 1.24[b] | 26.44 ± 2.72[bc] |
| AST (U/L) | 14.15 ± 1.26[b] | 294.52 ± 9.73[b] | 14.05 ± 1.82[b] | 24.10 ± 2.63[c] |
| LDH (U/L) | 140.74 ± 8.19[a] | 545.40 ± 12.76[a] | 142.13 ± 8.51[a] | 198.86 ± 9.71[a] |

Notes.

Values are expressed as (mean ± SE) and $n = 10$. ALT, alanine aminotransferase; AST, aspartate aminotransferase; and LDH, lactate dehydrogenase.
Means within the same row in each category (mean ± SE) carrying different letters are significant at $P \leq 0.05$, where the highest mean value has the symbol (a) and decreasing in values were assigned alphabetically.

**Table 4  The effects of Art/Q/Zn (30 mg/Kg) on inflammatory markers within male rats exposed to either Acy (500 mg/Kg) alone or in combination with Art/Q/Zn for 30 successive days.**

| Groups | IL-6 (Pg/g) | TNF-α (Pg/g) | CRP (mg/L) |
|---|---|---|---|
| Control | 3.40 ± 0.53[c] | 4.48 ± 0.10[cd] | 2.24 ± 0.21[d] |
| Acy (500 mg/Kg) | 49.70 ± 4.62[a] | 62.75 ± 4.54[a] | 96.10 ± 3.79[a] |
| Art/Q/Zn(30 mg/Kg) | 3.82 ± 0.73[c] | 3.80 ± 0.88[d] | 4.24 ± 0.97[c] |
| Acy + Art/Q/Zn | 6.43 ± 1.23[b] | 10.75 ± 1.39[b] | 6.51 ± 0.91[b] |

Notes.

Values were expressed as mean ± SE; $n = 10$. IL-6; interleukin-6, TNF-α; Tumor necrosis factor Alpha and CRP; C-Reactive protein. Means within the same row in each category (mean ± SE) carrying the different letters are significant values at $P \leq 0.05$, where the highest mean value has the symbol (a) and decreasing in values were assigned alphabetically.

**Table 5  The effect of Art/Q/Zn (30 mg/Kg) on the antioxidant status of male rats' liver tissues after treatment with Acy alone, Art/Q/Zn alone or their combined administration for 30 successive days.**

| Groups | MDA (nmoles of MDA /g) | CAT (nmol/g of protein/min) | SOD (U/g of protein) | GPx (nmol/g of protein/min) |
|---|---|---|---|---|
| Control | 4.13 ± 1.92[c] | 11.79 ± 1.47[a] | 16.59 ± 1.62[b] | 12.73 ± 1.49[a] |
| Acy (500 mg/Kg) | 53.42 ± 5.79[a] | 2.49 ± 0.86[c] | 5.47 ± 1.44[d] | 5.32 ± 1.48[d] |
| Art/Q/Zn (30 mg/Kg) | 4.40 ± 1.25[c] | 12.57 ± 1.81[a] | 17.76 ± 1.65[ab] | 12.95 ± 2.69[a] |
| Acy+ Art/Q/Zn | 14.13 ± 1.16[b] | 4.21 ± 0.95[b] | 13.89 ± 1.58[c] | 10.45 ± 1.98[b] |

Notes.

Values are expressed as (mean ± SE) ; $n = 10$. MDA; malondialdehyde, CAT; catalase, SOD; superoxide dismutase, GPx; glutathione peroxidase. Means within the same row in each category (mean ± SE) carrying the different letters are significant values at $P \leq 0.05$, where the highest mean value has the symbol (a) and decreasing in values were assigned alphabetically.

CRP, IL-6, and TNF-α levels were significantly elevated in the Acy treated group as compared with control group (Table 4). The CRP, IL-6, and TNF-α levels were markedly reduced in groups either treated with Art/Q/Zn alone or combined with Acy administration (Table 4).

Table 5 clarified that Acy afforded oxidative injury in the hepatic tissues, the level of the antioxidant enzymes were significantly declined due to treatment with Acy. Acy markedly declined CAT, SOD, and GPx activities. The Acy-group showed higher MDA levels than the control group, which is an indicator of an increment of oxidative injury. Meanwhile, administration of Art/Q/Zn afforded significant increase in the antioxidant enzymes CAT, SOD and GPx activities with reduction of MDA level (Table 5).

## Histological and ultrastructural examination of liver and lung tissues (histological, TEM, and immunostaining sections) with live sections clarifying variations in the structure of studied tissues

Following Acy administration, hepatic tissues displayed toxicity in the form of fatty change with considerable and elevated hepatocyte degeneration, as well as a congested portal vein with severe hemorrhage and infiltration of blood sinusoids by mononuclear inflammatory cells (Fig. 15). After the administration of Art/Q/Zn, the liver showed alleviation of hepatotoxicity with restoration of normal hepatic tissues (Fig. 15).

TEM examination of hepatic tissues showed normal structure in the control group, and Art/Q/Zn treated group with normal nucleus with clear nuclear boundaries, normal-sized mitochondria, and endoplasmic reticulum. Meanwhile, the treated group with Acy and Art/Q/Zn showed restoration of the normal hepatic structures with regular nuclear boundaries and significantly alleviated the hepatotoxicity (Fig. 16).

Immunostaining of the hepatic tissue sections showed high immunoreactivity for caspase-3 in the hepatic tissues of Acy treated group. Meanwhile, the immunoreactivity for caspase-3 was very weak in the Acy treated group and followed by Art/Q/Zn, as shown in (Fig. 17).

After administration of Acy and Art/Q/Zn, rat lung tissues showed pulmonary toxicity in the form of congested blood vessels with scattered aggregates of inflammatory cells, and appearance of granular debris of fibrin and congested blood vessels with areas of hemorrhage with non-specific inflammatory cells (Fig. 18). Administration of novel Art/Q/Zn after Acy improved the structure of lung tissues and showed alleviation of pulmonary fibrosis with restoration of normal structure of the pulmonary tissues (Fig. 18).

TEM of lung tissues showed normal structure in the control and the Art/Q/Zn treated groups with appearance of normal nucleus, basement lamina, alveolar sacculus, and normal Bronchioles. Meanwhile, the Acy treated group showed sizeable red blood cells that entirely blocking of the air sacs with appearance of small granules with detaching of most pulmonary tissues with pulmonary fibrosis. Meanwhile, normal pulmonary sacculus and alveolar epithelial cells were restored in Acy and Art/Q/Zn treated group with alleviation of the pulmonary fibrosis and hemorrhage (Fig. 19).

Immunostaining of the pulmonary tissues showed high immunoreactivity for caspase-3 in the group treated with Acy. Meanwhile, the immunoreactivity for caspase-3 was very weak in Acy treated group and followed by Art/Q/Zn, as shown in (Fig. 20).

Figure 21 demonstrated various morphological alterations in both lung and liver tissues of different treated groups. Lung tissues of Acy treated group as shown in Fig. 21A showed large lesions with congested tissues and bleeding. On the other hand, the lung tissues of Art/Q/Zn treated groups as shown in Fig. 21B are of normal appearance with a bright red color and without any lesions or any structural changes. Liver of Acy treated group as shown in Fig. 21 (C1, C2, and C3) showed hepatomegaly, dark oxidative color, and some lesions. Meanwhile, liver tissues of Art/Q/Zn treated groups as in Fig. 21 (D1, D2, and D3) showed bright normal sized hepatic tissues without any appeared lesions.

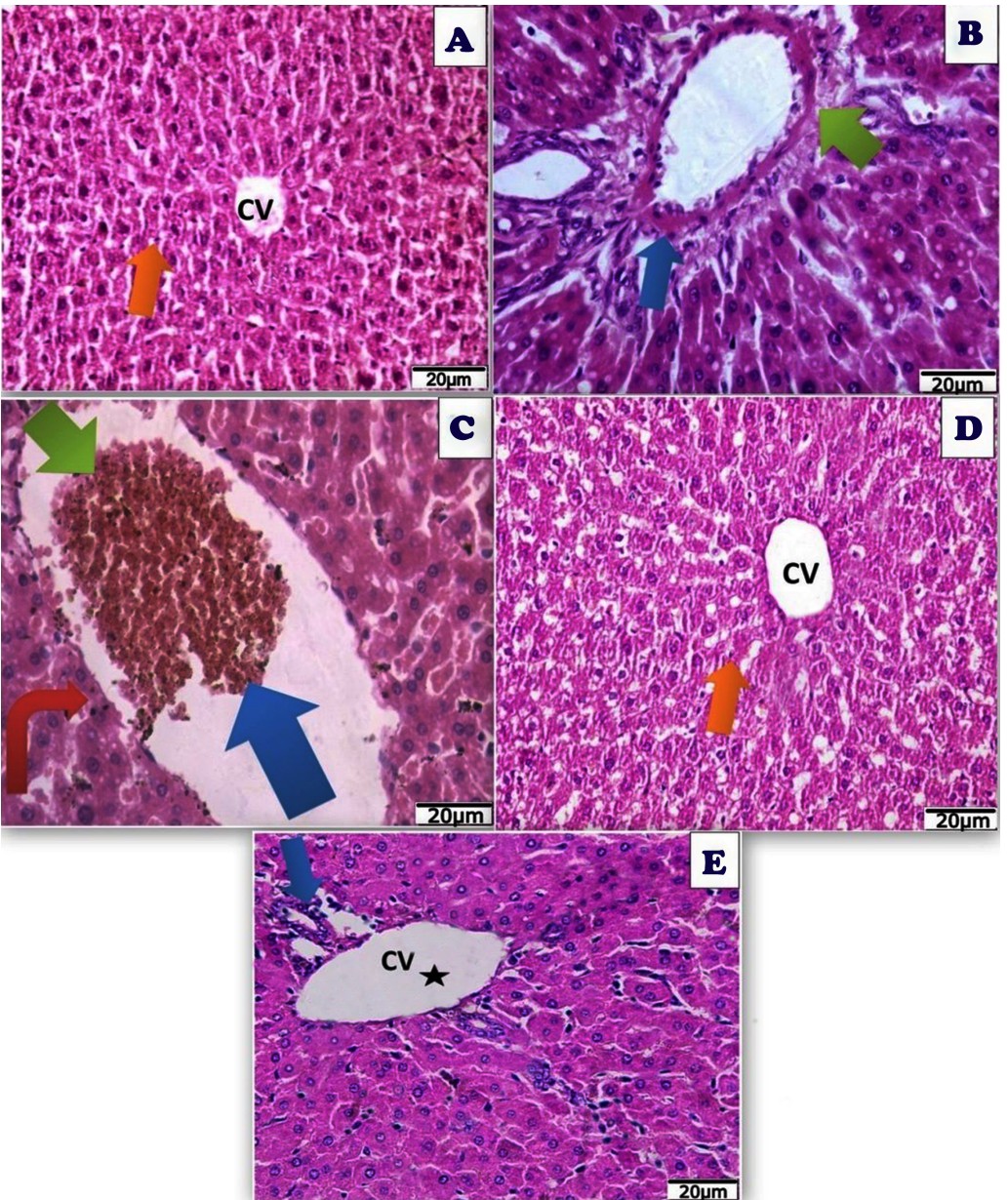

**Figure 15  Photomicrograph of cross sections of experimental rat liver.** A cross section of experimental rat liver showing: (A) control group showing normal hepatic structure with normal hepatocytes (orange arrow) and normal central vein (CV) (20 μm). (B) Acy treated group showing hepatotoxicity by hypertrophy of hepatocytes and increased eosinophilia, granular cytoplasm, vesicular nuclei and ballooning degeneration (blue arrow), dilatation of the portal vein with appearance of perivenular, periportal fibrosis (blue arrow), ductular reaction (green arrow) with accumulation of mononuclear inflammatory cells in the portal tract (interface hepatitis), focal necrosis in some hepatocytes with increased eosinophilia and nuclear disappearance (20 μm). (continued on next page...)

**Figure 15 (…continued)**
(C) A cross section of liver tissues after Acy administration showing severe toxicity with hypertrophy of the hepatocytes and elevated eosinophilia, granular cytoplasm with vesicular nuclei (red arrow), with hemorrhage in the central vein (blue arrow) and necrotic tissues (green arrow) and nuclear disappearance, accumulation of few mononuclear inflammatory cells in blood sinusoids (20 μm). (D) A cross section of liver tissues after (Art/Q/Zn) administration showing normal hepatic structure with normal central vein (CV) and normal hepatocytes (orange arrow) (20 μm). (E) A cross section of liver tissues after administration of Acy followed by the novel complex (Art/Q/Zn) with showing high restoration of the hepatic tissues in the form of very mild fatty change, with some inflammatory cells resulting from the recovery process (blue arrow) with normal central vein (black star) (20 μm).

## Effect of Art/Q/Zn on blood pressure levels

The control group showed standard heart rate with normal blood pressure recorded by systolic pressure 134.9 mmHG, diastolic pressure 108.7 mmHG and heart rate 263.2 beats/min (bpm) (Fig. 22A). Acy induced elevation in systolic and diastolic blood pressure by 188.20 mmHG and 162.02 mmHG, respectively with slightly higher heart rate recorded as 297.0 beats/min (bpm) than that of the control group (Fig. 22B). The Art/Q/Zn complex caused a significant reduction in systolic pressure in the control group. The Acy group recorded 102.05 mmHG as a systolic blood pressure and diastolic blood pressure as 82.94 mmHG after successive 30 days of treatment. Still, with a significant increment in heart rate by 379.78 beats/min (bpm) (Fig. 22C). However, treatment of male rats with a combination of Acy followed by the Art/Q/Zn complex induced a more pronounced reduction in systolic and diastolic blood pressures (by 134.95 and 263.54 mmHg) than in the Acy-only treated group with the lowering of heart rate recorded as (263.54 beats/min) (bpm) (Fig. 22D), All measurements for systolic and diastolic blood pressure with recording heart rate for all treated groups were carried out by digital blood pressure measurement system (NIBP250, BIOPAC systems, Inc., Goleta, CA, USA) and values were measured in rats after putting in the streamer (Fig. 22E). Systolic and diastolic values were shown in both (Fig. S2 and Table 6).

## Molecular docking assays

In the initial phase of the docking assay, the program was set up by redocking the novel formula (Art/Q/Zn) against two receptors. Angiotensin-converting enzyme-2 (ACE2). This receptor may generate a highly potent protective effects in the COVID-19 patients by reducing the severe respiratory symptoms risk, and $M^{pro}$ targets the main protease receptor for SARS-CoV-2. It is the the key and vital enzyme of coronaviruses and has a vital role in mediating the highly viral replication and transcription, as shown in Fig. 23A (v1-v6). Also, lipophilicity capacities were shown as most therapeutics essentially used specialized transport systems of the body, and instead of that mechanism, others tend to diffuse *via* the cellular membrane. To do so, therapeutics must be sufficiently lipophilic. So, if therapeutics showed lipophilicity affinity, this enables easy cellular diffusion, thus getting a much better effect of the therapeutics. Figure 23 showed high lipophilicity capacities of novel synthesized complex (Art/Q/Zn) with both SARS-CoV-2 receptors (ACE2) (Fig. 23B) and primary SARS-CoV-2 inhibitor ($M^{Pro}$) (Fig. 23C).

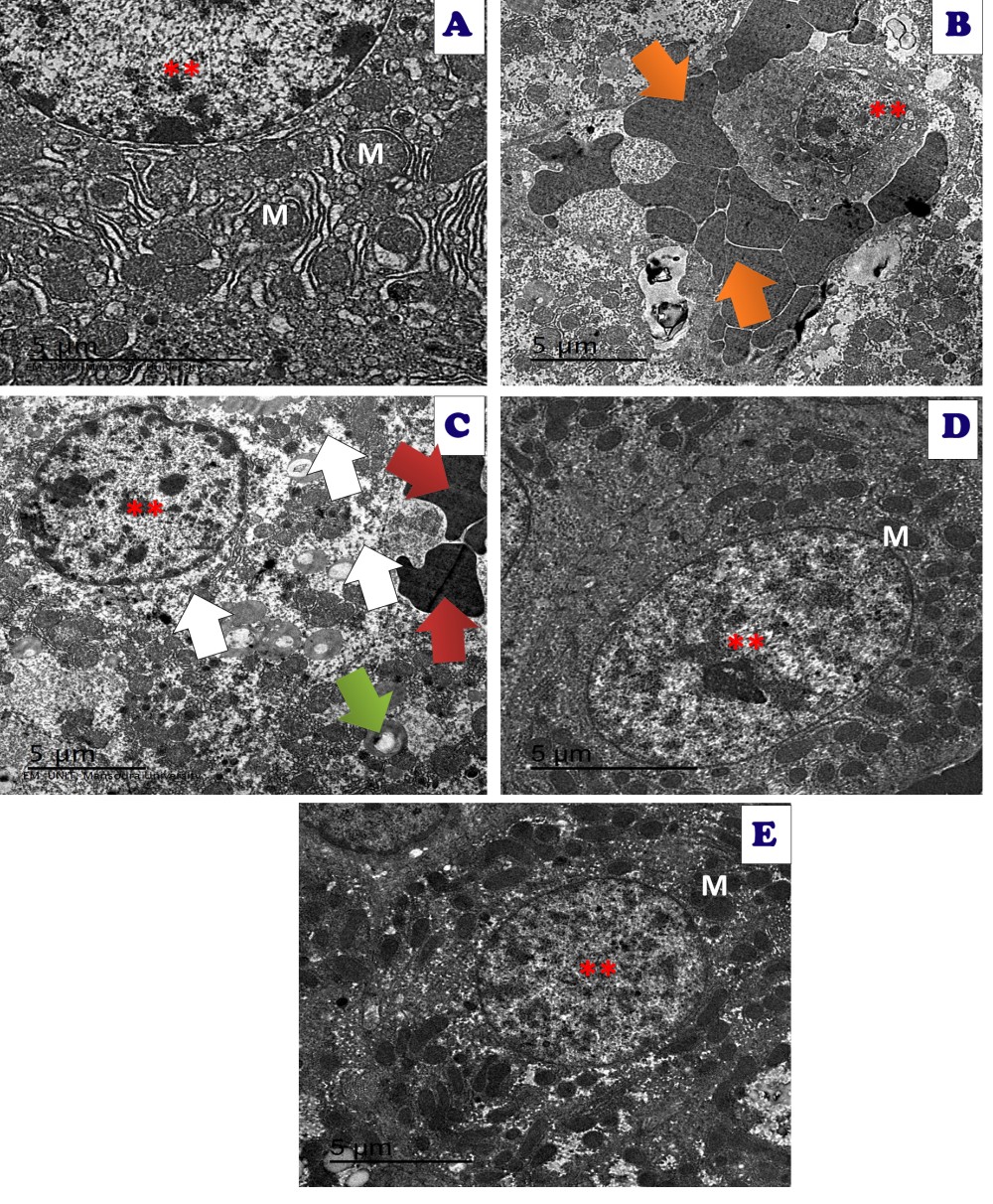

**Figure 16  An electron micrograph of the liver tissues of different treated groups.** An electron micrograph of the hepatic tissues of the control group: (A) which showed a normal hepatic structure with normal appearance of normal nucleus (**), normal mitochondria (M) and endoplasmic reticulum (ER) (scale bar = 5 μm). (B) Acy treated group showed large red blood cells (orange arrow) with large fat droplets appeared as large white droplets with appearance of pyknotic nuclei with very small size (**) (scale bar = 5 μm). (C) Acy treated group showed high fatty change (white arrow) with some destructed mitochondria (green arrow), hemorrhage with appearance of red blood cells (red arrow) and med sized nucleus (**) in addition to dysregulation of nuclear membrane (scale bar = 5 μm). (D) Art/Q/Zn treated group showing normal hepatic structures with enlarged nucleus with normal nuclear boundaries (**) and normal sized mitochondria (M) (scale bar = 5 μm). (E) Acy plus novel mixed ligand complex (Art/Q/Zn) showed restoration of most of the hepatic structure with normal sized nucleus (**) with normal sized mitochondria (M) and reducing of fatty change (scale bar = 5 μm).

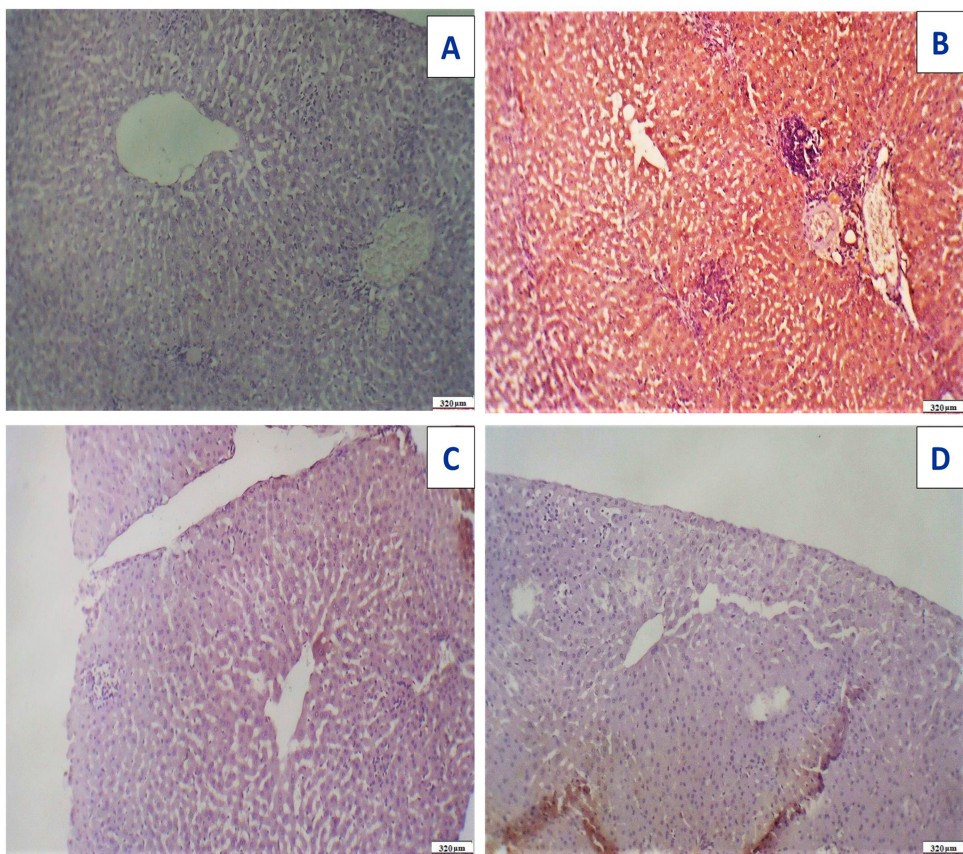

**Figure 17** **Immunostaining reactivity of the hepatic tissues.** (A) Control group: photomicrograph of a cross-section of the liver tissues showing negative caspase-3 immunostaining (-) negative immunostaining (320 µm). (B) Photomicrograph of a cross-section of the liver tissues given Acy showed highly marked hepatocyte immunostaining of caspase-3 indicating highly apoptosis and hepatotoxicity with inflammatory cells (++++) very immunostaining (320 µm). (C) Photomicrograph of a cross-section of the hepatic tissues after administration of novel complex of mixed ligand (Art/Q/Zn) showing negative cytoplasmic hepatocyte immunostaining of caspase-3 and induction of severe apoptosis and more inflammatory cells (-) negative immunostaining (320 µm). (D) Hepatic tissues of group treated with Acy and novel complex (Art/Q/Zn) showed very mild caspase-3 immunostaining and absence of inflammatory cells (+) weak immunostaining (320 µm) (DAB chromogen, Meyer's hematoxylin counterstain).

## Molecular dynamic simulation

The molecular dynamic (MD) simulation was performed for Art/Q/Zn at (100 ns) by using, a Package of Schrödinger (Desmond) Fig. 23A. Simulations were carried out to predict the ligand binding status in the physiological environment to mimic the expected binding of the novel complex formula (Art/Q/Zn) with the primary receptors of ACE2 and $M^{Pro}$ (The main protease of SARS-CoV-2) (Figs. 23B and 23C).

## Histogram and heat map analyses

Histograms were described in Fig. 24 (A1 and B1) and heat maps are shown in Fig. 24 (A2 and B2) for the SARS-CoV-2 ligand-protein of Art/Q/Zn novel complex with ACE2 and $M^{Pro}$, respectively, during the simulation time (100 ns).

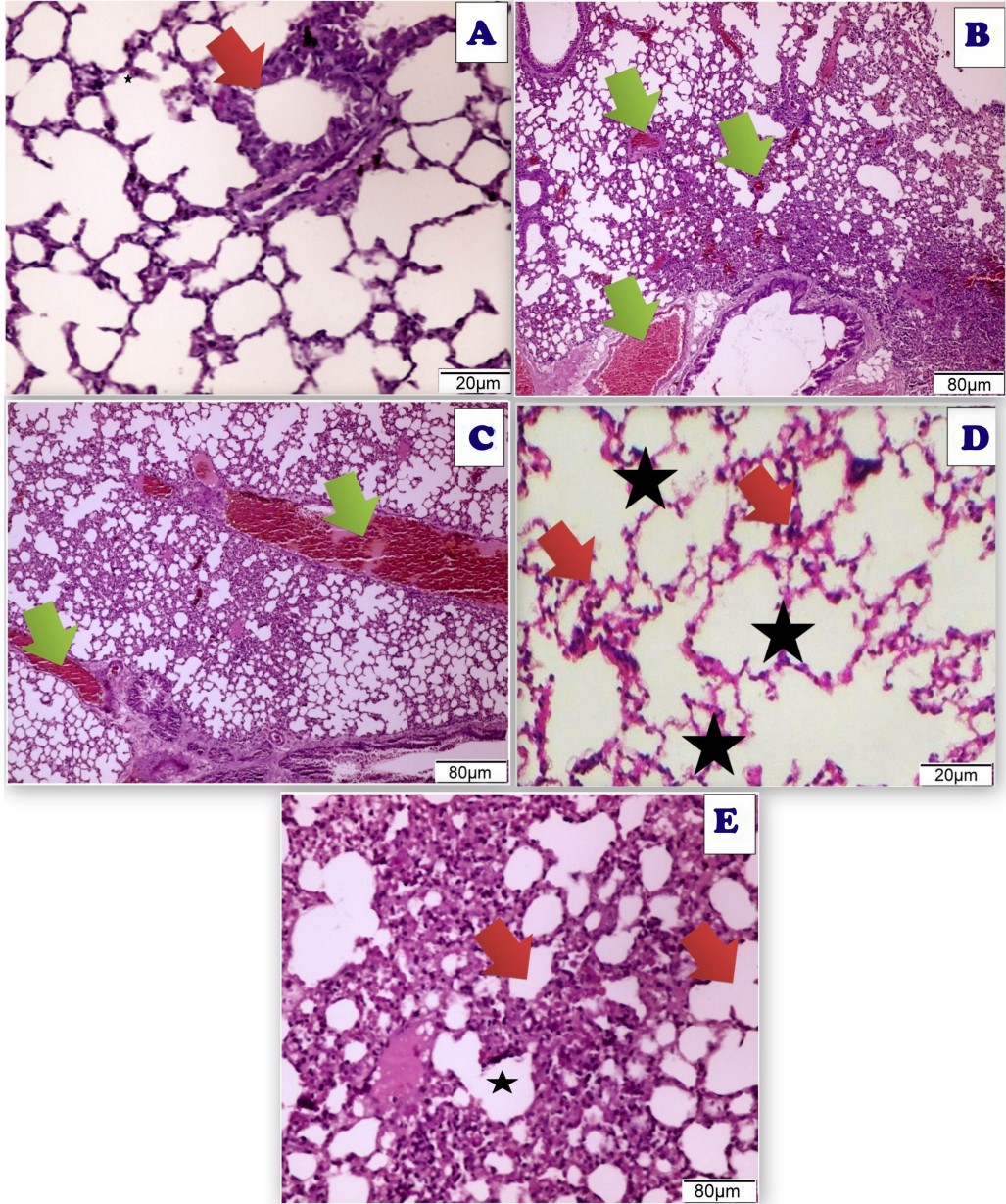

**Figure 18  Photomicrograph of cross sections of experimental rat lung.** (A) Control group showing the normal perivascular cells of the lung tissues (20 μm). (B) Acy treated group showing pulmonary toxicity showing congested blood vessels (green arrow) with scattered aggregates of inflammatory cells (80 μm). (C) Sections of rat lung treated with Acy administration that show severe toxicity in the form of granular debris of fibrin and congested blood vessels and areas of hemorrhage with non-specific inflammatory cells (80 μm). (D) A cross section of rat lung after novel complex (Art/Q/Zn) administration that show normal pulmonary sacs with regular boundaries and normal perivascular cells of the lung tissues (20 μm). (E) A cross section of rat lung after administration of Acy followed by the novel complex (Art/Q/Zn) that showing restoration of the lung tissues with very mild congestion of the alveolar lung tissues (80 μm).

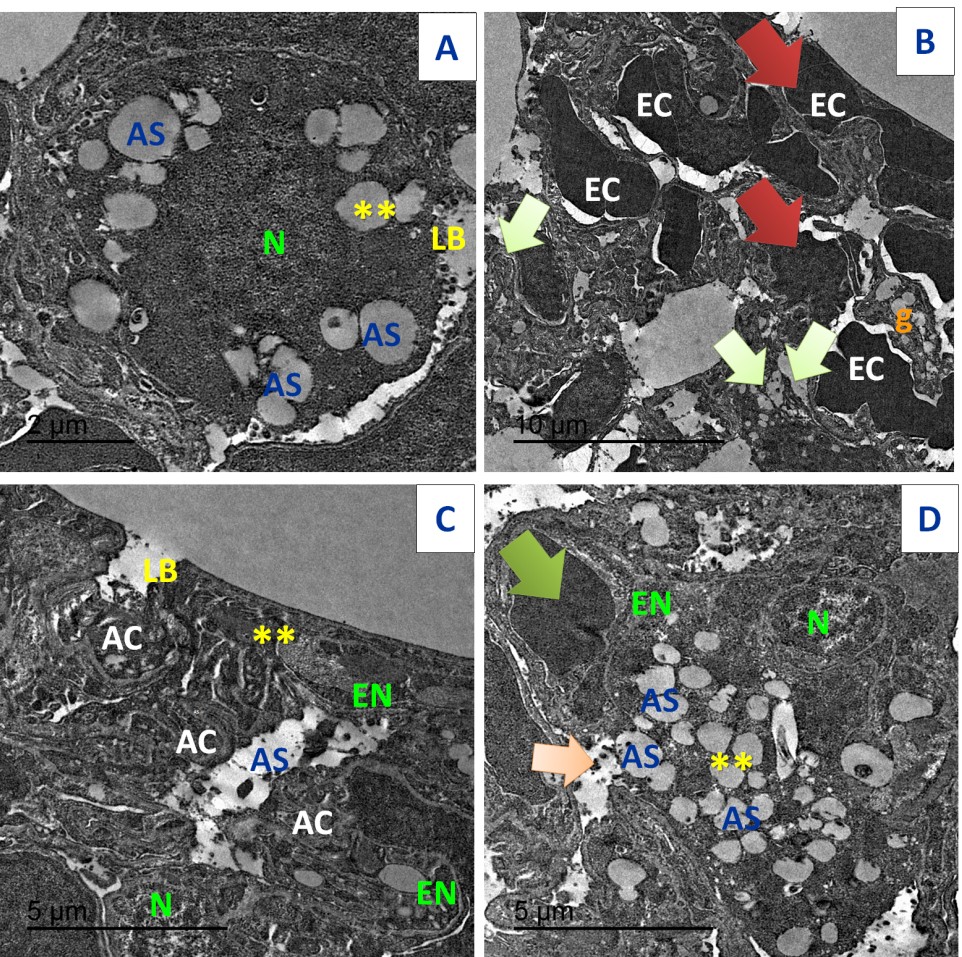

**Figure 19  TEM examination of the pulmonary tissues.** (A) showed normal pulmonary structures with normal appearance of the nucleus (N), basement lamina (LB) and alveolar sacculus (AS) and normal Bronchioles (**) (scale bar = 2 μm). (B) Acy treated group showed large red blood cells (EC) that block completely the air sacs (red arrow) with appearance of small granules (g) with detaching of most pulmonary tissues with pulmonary fibrosis (green arrow) (scale bar = 10 μm). (C) Art/Q/Zn group showing normal pulmonary structures with normal nucleus (N) beside normal nuclear boundaries and normal Bronchioles (**), normal Alveolar epithelial cells (AC), alveolar sacculus (AS) and normal basal lamina (LB) (scale bar = 5 μm). (D) Acy plus novel mixed ligand complex (Art/Q/Zn) showed restoration of most of the pulmonary structure with normal sized nucleus (N), normal alveolar epithelial cells (AC), alveolar sacculus (AS) and opening of air sacs (**) with mild appearance of granules residues (light orange arrow) and red blood cells (green arrow) (scale bar = 5 μm).

## Ligand properties

Regarding the novel complex formula Art/Q/Zn with ACE2 Fig. 24 (A3), GLN-72 contributed mainly about 99%, besides GLN-59 contributed mainly about 49%, followed by LYS-51, LEU-54 and PHE-55 contributed mainly as follows, respectively (41,66, 41,38 and 31%) of the interactions as H-bonding; However, ILE-61, VAL-93, ILE-19 and MET-62 formed essentially the hydrophobic interactions. LYS-51, LEU-54, PHE-55, GLN-59, and GLN-72 were the most members contributing to the water-bridges, with no record of ionic

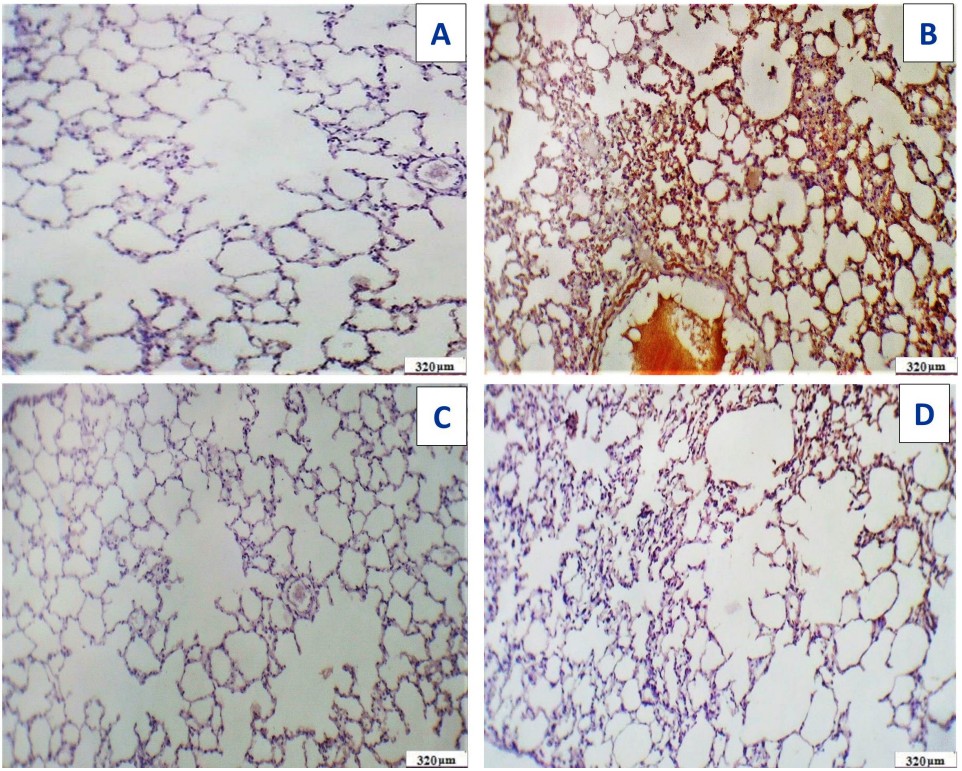

**Figure 20 Immunostaining reactivity of the pulmonary tissues.** (A) Section of the pulmonary tissues of control group showing (-ve) caspase-3 immunostaining (320 μm). (B) Cross-section of the pulmonary tissues given Acy showed significant immunostaining of caspase-3 indicating highly apoptosis (++++) very immunostaining (320 μm). (C) Cross-section of the pulmonary tissues after administration of novel complex of mixed ligand (Art/Q/Zn) showing negative cytoplasmic hepatocyte immunostaining of caspase-3 and induction of severe apoptosis and more inflammatory cells (-) negative immunostaining (320 μm). (D) Section of the pulmonary tissues treated with Acy and novel complex (Art/Q/Zn) showed very mild immunostaining for caspase-3 and absence of inflammatory cells (+) weak immunostaining (320 μm) (DAB chromogen, Meyer's hematoxylin counterstain).

bonds. GLN-72 was the most essential participating amino acid in the interactions through hydrogen bonds.

By analyzing the docking results for our synthesized complex against ACE2 and $M^{pro}$ receptors of SARS-CoV-2, we can conclude that the protein-ligand contact is as follows: the reduced cocrystallised formula formed hydrogen bonds in case of ACE2 with GLN-18, ILE-19, GLN-24, LYS-51, LEU-54, PHE-55, LEU-57, GLY-58, GLN-59, ILE-61, MET-62, TYR-67, GLN-72, VAL-75, VAL-93, LYS-94, HIS-96 and TYR-100 incase of ACE2 receptor (Fig. 24 (A3)) and PHE-8, LYS-102, VAL-104, ARG-105, ILE-106, GLN-107, GLY-109, GLN-110, THR-111, ASN-151, ILE-152, ASP-153, TYR-154, CYS-156, SER-158, CYS-160, ASN-203, ASP-248, PHE-294, THR-292, ASP-295, ARG-298 and GLN-306 incase of $M^{Pro}$ with a very high rate of activity and the novel complex Art/Q/Zn showed a binding interaction energy as shown in Fig. 24 (B3).

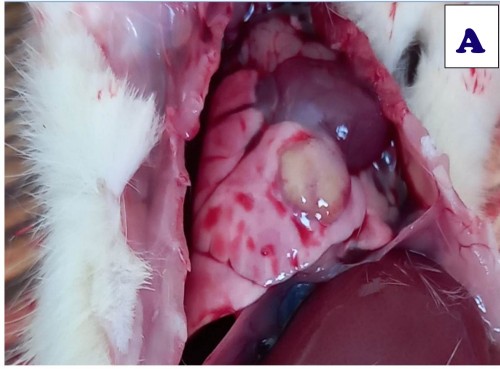
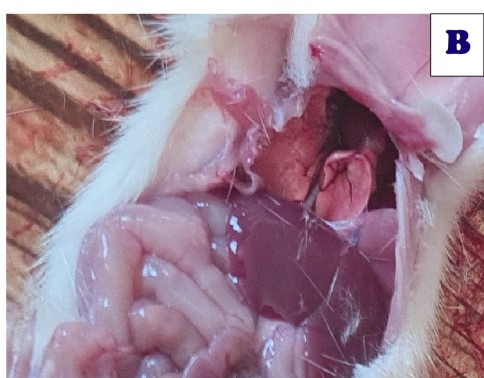
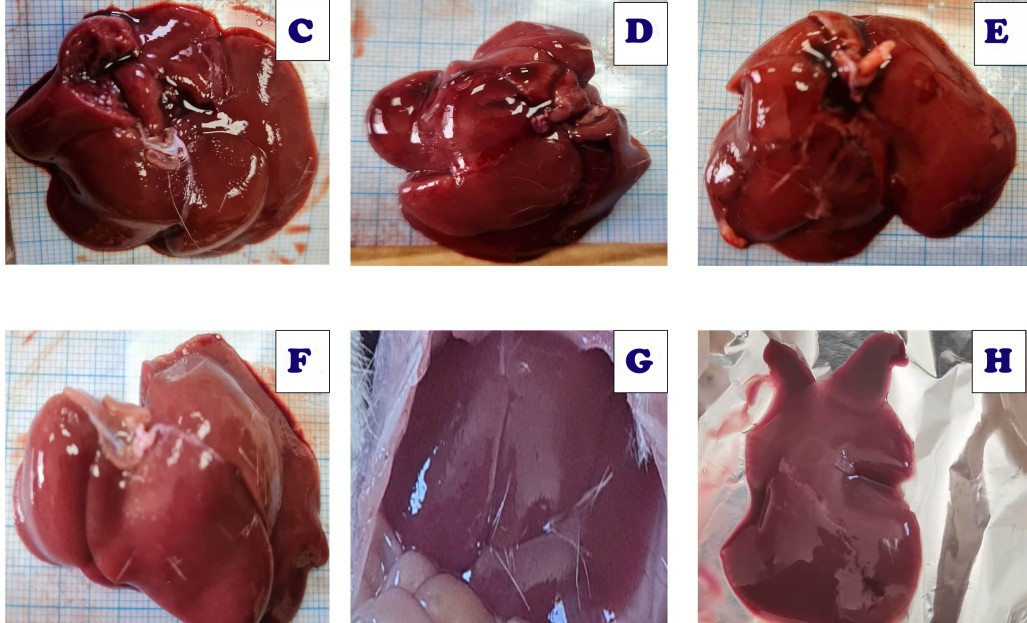

**Figure 21 Live sections of both liver and lung tissues of either Acy and Art/Q/Zn treated groups.** (A) Lung tissues of Acy group. (B) Lung tissues of ART/Q/Zn group. (C, D, and E) Hepatic tissues of Acy treated group. (F, G, and H) Hepatic tissues of ART/Q/Zn treated group.

Additionally, the SARS-CoV-2 ligand-protein of the novel complex Art/Q/Zn with the receptor M$^{\text{pro}}$ during the simulation time (100 ns) as described in Fig. 24 (B3). Regarding the novel complex Art/Q/Zn with M$^{\text{Pro}}$, GLN-10 contributed about 99%, besides PHE-8 contributed only about 60%, followed by ASP-298 and THR-111, which contributed as follows, respectively (58 and 57%) of the interactions as H-bonding. However, PHE 294 and LYS-102 formed mostly the hydrophobic interactions. ASP-248, THR-111, CYS-156 and GLN-110 were the main members contributing to the water-bridges, no record of ionic bonds in both ILE-152 and ARG-298. Obviously, GLN-10 was the most participating amino acid in the interactions through hydrogen bonds.

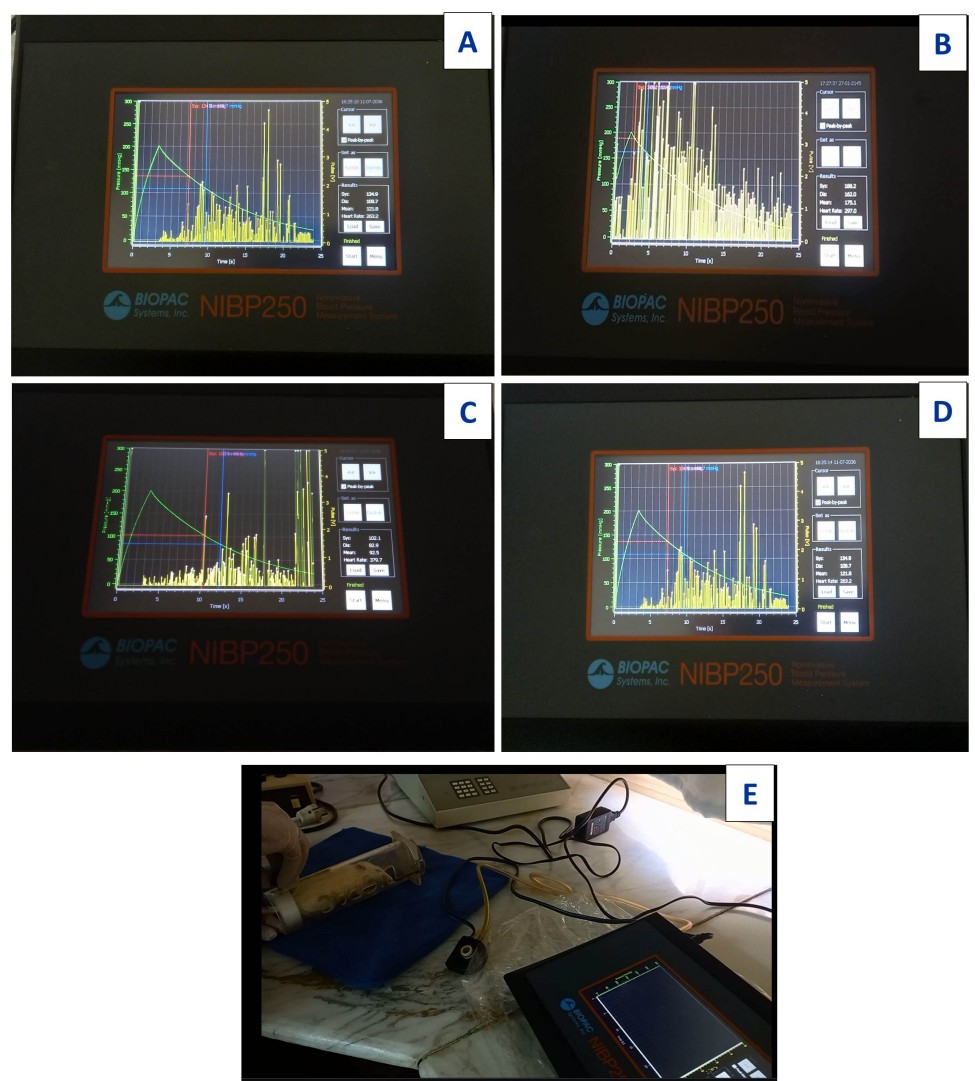

**Figure 22 Images for digital measurement blood pressure system (NIBP250), BIOPAC systems, inc.**
Systolic and diastolic blood pressure in Acy and /or Art/Q/Zn treated group. (A) Systolic, diastolic and
heart rate measurements of control group. (B) Systolic,Diastolic and heart rate measurements of Acy
group. (C) Systolic, diastolic and heart rate measurements of Art/Q/Zn group. (D) Systolic, diastolic
and heart rate measurements of Acy +Art/Q/Zn group. (E) Digital blood pressure measurement system
(NIBP250), BIOPAC systems, Inc., and values were measured in rats *via* streamer.

## Root mean square deviation analysis

The root mean square deviation (RMSD) values of C$\alpha$ atoms were evaluated the novel
complex Art/Q/Zn to monitor the effect of this novel complex on the conformational the
high stability of ACE2 and M$^{pro}$ receptors during the current simulations—the results as
seen in Fig. 24 (A4 and B4). The fluctuation of the proteins was within acceptable variation
with RMSD values that were less than 2.00° A in the case of ACE2 and less than 3.00° A in
the case of M$^{Pro}$, indicating the stability of the protein conformation.

**Table 6** Assessment of Art/Q/Zn (30 mg/Kg) on blood pressure (systolic, diastolic and heart rate) levels of male rats treated with Acy (500 mg/Kg) alone or in combination with Art/Q/Zn for successive 30 days.

| Groups | Systolic (mmHg) | Diastolic (mmHG) | Heart rate (Pulse/min) |
|---|---|---|---|
| Control | $134.90 \pm 4.02^b$ | $108.74 \pm 2.58^b$ | $263.21 \pm 5.69^c$ |
| Acy | $188.20 \pm 3.64^a$ | $162.02 \pm 5.25^a$ | $297.08 \pm 4.68^b$ |
| Art/Q/Zn | $102.05 \pm 1.36^c$ | $82.94 \pm 5.69^c$ | $379.78 \pm 5.68^a$ |
| Acy + Art/Q/Zn | $134.95 \pm 5.02^b$ | $108.75 \pm 6.25^b$ | $263.54 \pm 3.68^c$ |

**Notes.**
Means within the same row in each category mean ± SE carrying different letters are significant at $P \leq 0.05$, where the highest mean value has the symbol (a) and decreasing in values were assigned alphabetically.

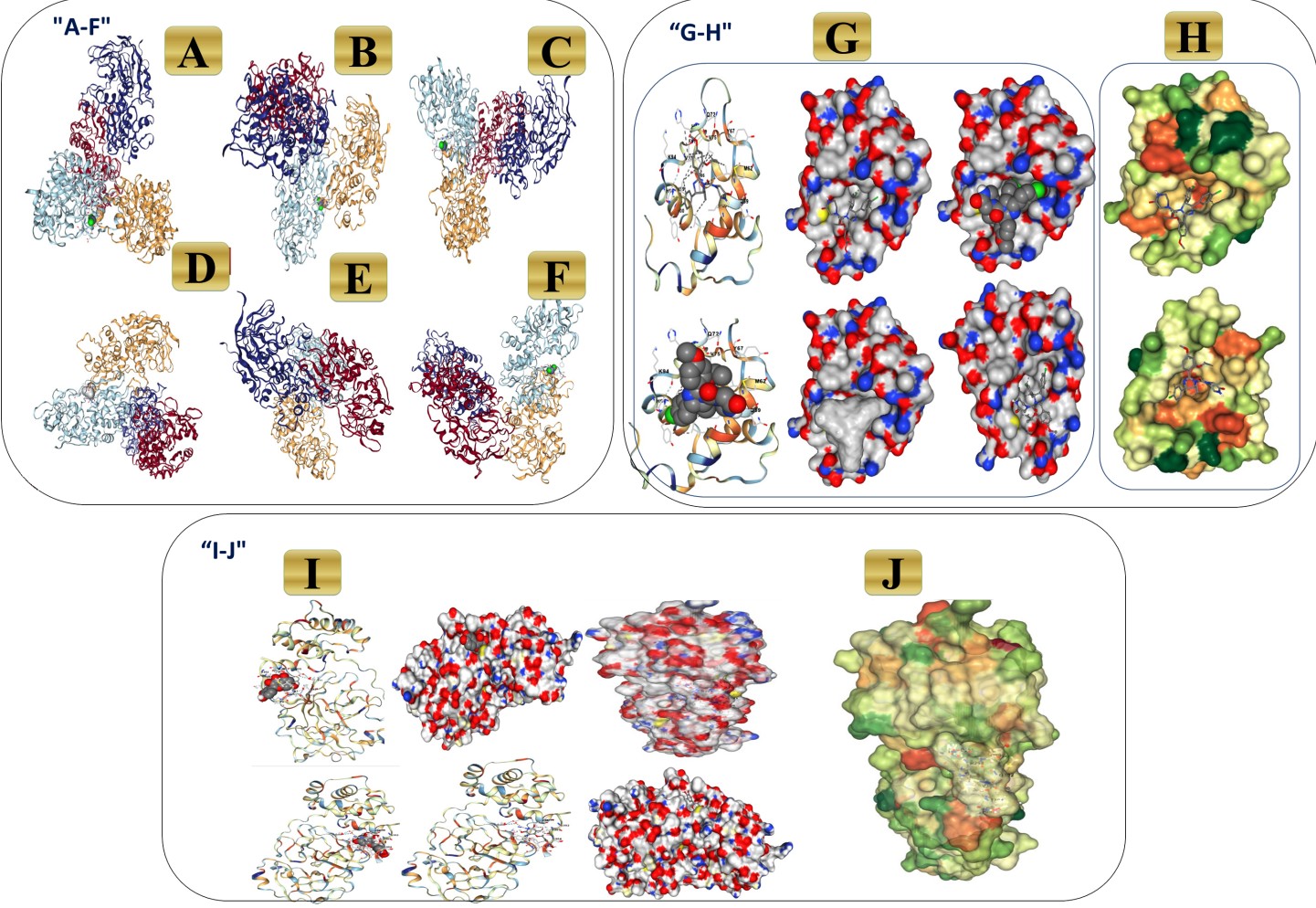

**Figure 23** **3D binding interaction of novel complex formula (Art/Q/Zn) with ACE2 and M^Pro receptors.** (A–F) 3D binding interaction of (Art/Q/Zn). (G–H) Simulation of the novel complex of (Art/Q/Zn) with ACE2 receptor. (G) 3D Simulation of Art/Q/Zn with ACE2 (H) Lipophilicity of Art/Q/Zn with ACE2 (I–J) Simulation of novel complex of (Art/Q/Zn) with M^Pro receptor. (I) 3D Simulation of Art/Q/Zn with M^Pro (J) Lipophilicity of Art/Q/Zn with M^Pro.

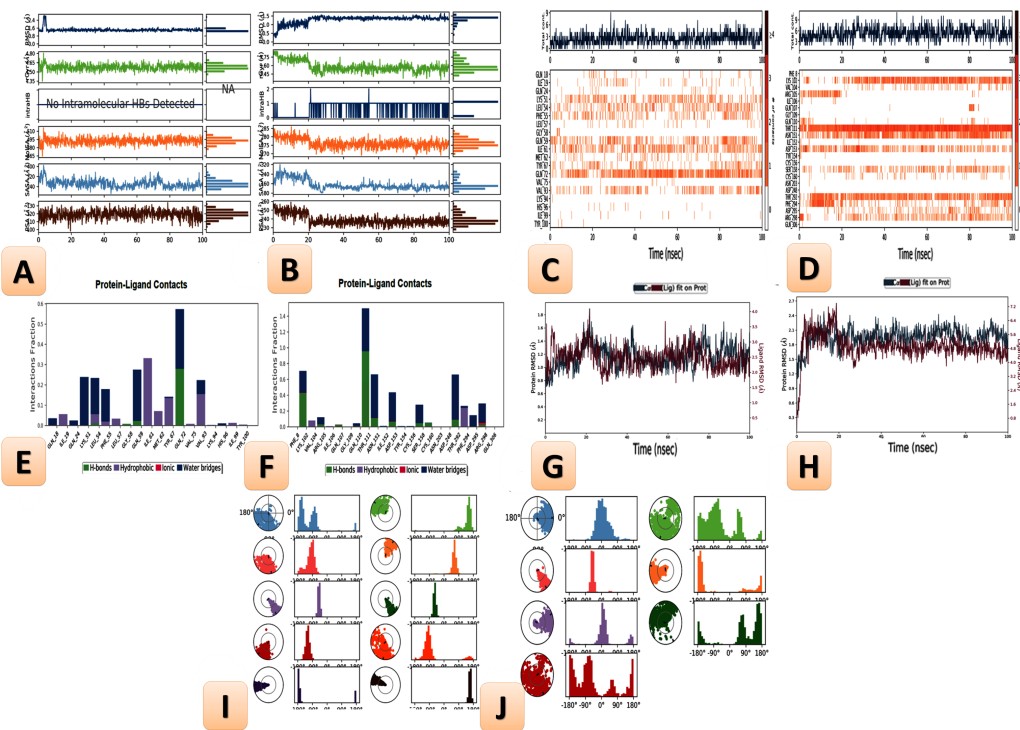

**Figure 24 Ligand properties, heat maps for SARS-CoV-2, L-torsion and RMSD of the tested mixed ligand (Art/Q/Zn) towards SARS-CoV-2 receptors during (100 ns), RMSD of ACE2, M^Pro and RMSD of Cα atoms of the complex ligand.** Molecular dynamics simulation to predict the ligand binding status of the SARS-CoV-2 ligand-protein for novel complex Art/Q/Zn with 1^ry receptor ACE2 and M^Pro (Main protease of SARS-CoV-2) (Simulation time 100 ns). (A, B) Histograms of molecular dynamics simulation. (C, D) Heat maps of molecular dynamics simulation. (E, F) Ligand properties of molecular dynamics simulation. (G, H) root mean square deviation (RMSD) analysis of molecular dynamics simulation. (I, J) Fluctuation and torsion of molecular dynamics simulation.

The features of the ligand, including the RMSD, radius of gyration (rGyr), solvent accessible surface area (SASA), the intramolecular hydrogen bond (intraHB), molecular surface area (MolSA), and polar surface area (PSA), all these features are shown in Fig. 24 (A4 and B4). Other ligand characteristics are reported in the root mean square deviation (RMSD).

The RMSD and rGyr for Art/Q/Zn novel complex with ACE2 were determined to be within the range of (0.7–1.9) Å. Also, no intraHB bands were observed during the 100 ns of simulation, and the MolSA range was within (65–512 Å2). Additionally, the SASA was within the (35–400 Å2). Additionally, its PSA range was within the range of 10 and 130 Å2 (Fig. 24) (A4).

The RMSD and rGyr for Art/Q/Zn novel complex with M^Pro were observed within the (0.3–2.6)Å range, respectively. Also, intraHB with bands were observed range to~2 during the 100 ns of the current simulation, and the MolSA range was within (70–285 Å2). Furthermore, the SASA was within the range of (80–320 Å2). Moreover, its PSA range was between 25 and 260 Å2 (Fig. 24) (B4).

The ligand properties showed fluctuation and torsion at the beginning of the simulation Fig. 24 (A5 and B5) before reaching the equilibrium state, indicating the high stability of the Art/Q/Zn complex to the main active site of the SARS-CoV-2 main protease active sites.

## DISCUSSION

The COVID-19 pandemic posed a serious global challenge that has altered global economic health. Thus, we need to check the effectiveness of recently discovered active compounds and combine them with drugs to obtain a novel formula to increase their therapeutic effectiveness against resistant strains of bacteria and viruses.

This study aimed to study the structure and anitviral activity of a novel synthesized metal complex Art/Q/Zn. The $IC_{50}$; $CC_{50}$; antioxidant capacities; and physiological, histological, and ultrastructural effects of this complex in the liver and lung against Acy-induced toxicity were tested *in vivo* in male rats. Our findings confirmed the novelty of Art/Q/Zn chemical structure and confirmed its high activity against the SARS-CoV-2. The novel complex Art/Q/Zn also exhibited high antioxidant activity against oxidative stress induced by Acy and improved the physiological functions of both the liver and lung. It also alleviated the structural alterations induced by Acy, especially in pulmonary tissues. It also restored the normal alveolar sacs and hepatic architecture and expressed negative-to-mild immunoreactivity against Caspase-3 immunostaining and declining inflammatory markers.

Additionally, the novel complex helped in counteracting the Acy-induced increase in blood pressure, highlighting its potential activity in restoring renal physiological functions. These findings also confirmed that the complex has a positive effect on ACE2 receptor, which is attached to the cellular membrane of the intestines, kidney, testis, gallbladder, and heart or in a soluble form and plays a vital role in maintaining the blood pressure. The complex helps ADAM17 in cleaving its extracellular domain to create soluble ACE2. Soluble ACE2 lowers blood pressure by catalyzing the hydrolysis of angiotensin II, a vasoconstrictor peptide, into angiotensin (1-7), a vasodilator (*Sanna et al.,2016*; *Grant, 1991*).This binds to MasR receptors, causing vasodilation and decreasing blood pressure (*Sanna et al., 2016*). Its potential to reduce blood pressure makes it a promising drug for treating hypertension and cardiovascular diseases (*Hamza et al., 2021*; *Grant, 1991*). Thus, this receptor plays a vital role in treating hypertension and maintaining renal function and its reabsorption abilities.

Art, along with other active compounds or metals, can be used to elevate the therapeutic effectiveness of this complex and inhibit the development of drug resistance. Therefore, it is crucial to develop a strategy to treat pandemic diseases. *Artemisia annua L.* is considered a safe and effective agent for treating hyperlipidemia, malaria, and other recorded inflammatory diseases. Additionally, this plant has antimicrobial and antiviral activities (*Iftekhar et al., 2022*).

SARS-CoV-2 mainly affects respiratory functions and leads to the development of acute respiratory distress syndrome. Therefore, the development of vaccines and alternative medicine that can alleviate the severe COVI-19 pandemic symptoms is garnering attention.

The finding of the current study greatly supported that our novel synthesized complex Art/Q/Zn has a remarkable ability as a potent anti-viral agent against the pandemic virus "SARS-CoV-2" and also prohibited the antioxidant capacities, anti-hepatotoxicity, and anti-pulmonary toxicity against toxicity induced by ACy in male rats and the obtained results are in excellent accordance with the previous results which demonstrated that clinical study indicated that treatment with one derivative of Artemisia markedly shortens the duration of the hospital stay for COVID-19 patients and decline their symptoms in China (*Karamoddini et al., 2017*; *Uzun & Toptas, 2020*).

The key strength of this study was the comprehensive chemical characterization of the novel Art/Q/Zn complex and confirmation of its chemical properties, behavior, and surface structure. This is the first study to reveal the high anti- SARS-Cov-2 activity of the novel complex without cytotoxicity. It has high antioxidant and antihepatotoxic effects against Acy, which induces hepatotoxicity and oxidative stress upon cooking food at high temperatures. This is a major challenge that threatens global health that can be addressed using this novel complex, which confers high antioxidant activity that can combat oxidative damage accompanied by severe inflammation, the current silent killer.

The findings extend our knowledge of the mechanism of action of our novel Art/Q/Zn complex against SARS-CoV-2. As for artemisinin, SARS-CoV-2 is (+ve) single-stranded RNA virus; it has four essential proteins (spike, envelope, membrane and nucleocapsid). SARS-CoV-2 invades host cells *via* two receptors: either angiotensin-converting enzyme 2 (ACE2) and/or CD147. The S protein on SARS-CoV-2 virus binds to ACE2 or CD147 on the cellular host, mediating viral invasion to the host cells and its dissemination to other cells and, thus, inhibition of expression of the two cellular receptors , and binding with Art/Q/Zn that will lead to inhibition of the viral entry, and subsequent excessive viral replication, thereby mitigating SARS-CoV-2 infection (*Kapepula et al., 2020*; *Uzun & Toptas, 2020*).

IIt is known that two active components of Artemisia (either artemisinin or artesunate) can retard the replication of the HCV, which is a positive sense virus and so is similar to the SARS-CoV-2 virus single-stranded RNA virus (*Karamoddini et al., 2017*). Additionally, we confirmed in the current study the anti-viral activity of our synthesized novel complex as it recorded high viral inhibition (IC50 equal 10.14 µg/ml) that can inhibit pandemic virus "SARS-CoV-2" viral invasion, replication and then reduce oxidative stress and thus eventually decline the inflammation during the pandemic of COVID-19 and these finding highlight the potential insights into the role of our novel synthesized complex in fighting against SARS-CoV-2 and elevate community health and immune system.

The current findings illustrate the mechanism underlying the anti–SARS-CoV-2 activity of the complex Art/Q/Zn, as shown in Fig. 25. Both ACE2 and CD147 in the host cells are the major SARS-CoV-2 receptors (*Kapepula et al., 2020*; *Ulrich & Pillat, 2020a*; *Shereen et al., 2020*). The S protein of SARS-CoV-2 can bind to the target cellular receptor, either ACE2 or CD147, and enter the host cells, where the virus can replicate (*Ulrich & Pillat, 2020b*; *Kaptein et al., 2006*). Thus, the main target mechanism is the ability of our synthesised

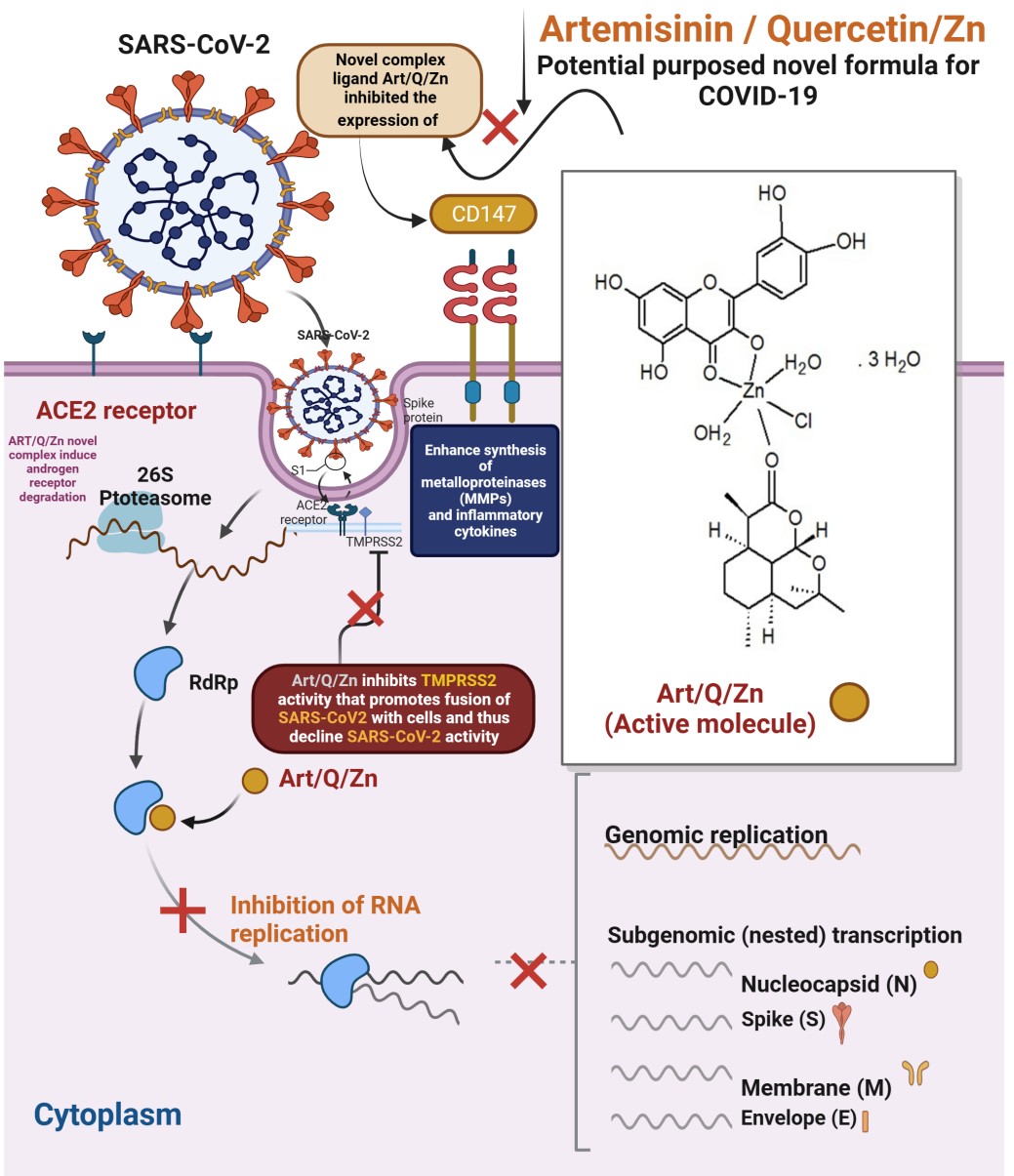

**Figure 25** **Proposed mechanism of antiviral activity of Art/Q/Zn.**

novel complex Art/Q/Zn to decrease cellular expression of ACE2, offering high protection against COVID-19 (*Kaptein et al., 2006*).

During its entry into the cellular host, SARS-CoV-2 requires cellular proteases, such as trypsin-like proteases (TMPRSS2) that enhance the cleavage of the S protein to enhance fusion of the host cell membranes and viral cellular membrane (*Wang et al., 2020*; *Elkhodary, 2020*; *Augustin et al., 2020*; *Hoffmann et al., 2020*).

Previously, Art has been indicated to have high activity against malaria and other hepatic viruses that cause various hepatic diseases. Art has an effective antiviral activity against

different viruses such as HIV, Ebola, and many coronaviruses (*Augustin et al., 2020*). Recently, the effectiveness of different medicinal plants, such as Art, against COVID-19 has been studied (*Hoffmann et al., 2020*). The current results revealed the ameliorative effect of the complex against lung congestion, which may be attributed to an Art derivative called Artesunate that inhibits the invasion of cancer cells by inducing the expression of essential cellular proteases, as previously demonstrated (*D'alessandro et al., 2020*; *Kalalagh et al., 2022*; *Rasheed et al., 2010*).

These findings confirmed the efficacy of the novel formula Art/Q/Zn in restoring pulmonary tissues because of the ameliorative effect of Art. Moreover, the synergistic effect of Q and Zn improves lung functions probably because of the antiviral activity of Zn, which improves the respiratory function affected by SARS-CoV-2. Thus, this complex may be a promising antiviral and antioxidant agent.

*Gendrot et al. (2020)* suggested that Art may prevent the onset of SARS-CoV-2 infection through the inhibition of the binding of SARS-CoV-2 S protein to ACE2, the main binding cellular receptor, thereby activating natural killer cells (*Rolta et al., 2020*). *Efferh (2018)* confirmed that Art may bind to E, N, 3CL$^{Pro}$, and S protein. The main biological function of Art may be attributed to its ability to inhibit the functions of these viral proteins.

The major health problem during the COVID-19 pandemic was the cytokine storm and elevation of inflammatory marker levels, which induced oxidative stress and damaged most human organs (*Krishna et al., 2015*). *Badraoui et al. (2022)* reported that Art extracts are safe, confirming the current results. These findings encourage the use of Art that can be administered *via* oral route or inhalation to capture the SARS-CoV-2–infected cells that form large colonies in the respiratory tract.

An additional advantage of Art/Q/Zn is that Art can bind to the target 6LU7, conferring stability to the formed complex and enhancing its inhibitory effect against the SARS-CoV-2 for an extended time.

Our obtained results are supported by those of a previous study (*Badraoui et al., 2022*) revealed that active compounds of Artemisinin exhibited promising properties, explaining the potent antioxidant, protective, and health-beneficial effects of Art.

Art, as reported by *Ben Nasr & Badraoui (2022)*, interacted in different ways with ACE2, and this confirmed the potential antiviral effect of Art and showed that this action was enhanced after complexation with Q/Zn.

As previously reported, Quercetin (Q) is considered a naturally occurring flavonoid that shows multi-beneficial biological effects. Moreover, Q administration minimized the severe oxidative damage. The supplementation of Q plays an important role as a protective agent by regulating the inflammatory responses (especially those resulting from viral infections) (*Refat et al., 2021*).

The beneficial effects of micronutrients in enhancing the immune response have earned more attention—particularly micronutrients such as Zn. Besides, Zn is vital in treating pulmonary diseases and subsequent dysfunction. In a previous analysis study, authors indicated that supplementation of Zn inhibited pneumonia. Meanwhile, a low dietary intake of Zn could considerably decline the human main resistance against infections (*Ben Nasr & Badraoui, 2022*).

Therefore, our novel synthesized complex Art/Q/Zn declines the activity of these metalloenzymes and thus could be essential for controlling the movement of severe infection of SARS-CoV-2.

Previous studies reported that androgens can up-regulate the expression of proteases such as TMPRSS2 protein and ACE2. So, we concluded that Art could induce androgen receptor degradation through the 26S proteasome. Therefore, Art might inhibit the infection of SARS-CoV-2 by limiting the expression of ACE-2 receptors or TMPRSS2 in host cells (*Iftekhar et al., 2022*).

CD147 can elevate the synthesis of metalloproteinases in cell–matrix and inflammatory cytokines. Art previously inhibited the expression of CD147 in human host cells. Additionally, Art strongly blocked CD147 expression. Therefore, Art might be highly effective in controlling and retarding the cytokine storm and thus inhibiting the infections by SARS-CoV-2 virus (*Iftekhar et al., 2022*).

In regard to the high capacities of Art, the addition of Art greatly enhanced the beneficial properties of Q/Zn complex, which possess antidiabetic, anti-infammatory, and antioxidant activities that can be used with stem cell therapy, and our previous study reported that Q/Zn has great benefits for lung tissues and in the alleviation of inflammation caused by diabetes mellitus induction (*Refat et al., 2021*). This observation was supported and enhanced by the results of complete chemical charaterization of the novel complex of the current study Art/Q/Zn complex. *Maret (2017)* and *Olechnowicz et al. (2018)* reported that Zn can upregulate the expression of inflammatory cytokines, play a key role in reducing the incidence of metabolic syndromes, and may suppress inflammation. Zn is an essential micronutrient that elevates the activities of antioxidant enzymes, which scavenge free reactive radicals and improve oxidative injury.

In complete agreement with the obtained results, Art/Q/Zn has improved the antioxidant capacities of male rats. As revealed before (*Refat et al., 2021*), Q/Zn which is a another correct separate and different complex than Art/Q/Zn, but this Q/Zn formula interacts with SOD and CAT enzymes. It enhances their antioxidant capacities in scavenging free radicals and thus decreasing oxidative injury. Also, Q/Zn complex previously ameliorated pancreatic and pulmonary both histological and ultra-structures, and this adds more strength point to the current data that the novel synthesized mixed ligand of Art/Q/Zn significantly reduced both hepatic and pulmonary structures and elevated antioxidant enzymes SOD, CAT and GSH with declining the final marker of lipid peroxidation MDA, and thus alleviation of oxidative injury by great scavenging capacities to the free radicals induced by Acy. Administration Art/Q/Zn was more potent in alleviation of the oxidative injury by higher effect than previously induced by Q/Zn.

The other bright side of the current study is the ability of the novel synthesized complex Art/Q/Zn to alleviate oxidative stress, hepatotoxicity, pulmonary toxicity, and both structural alterations induced by Acy, as these results are consistent with data obtained in previous study of *Hamza, Al-Motaani & Malik (2019)* as they confirmed that Acy caused many pathological alterations and cellular injury occur due to the imbalance between antioxidant enzymes in tissues and thus the effect of Art/Q/Zn was greatly noticed in the current study in alleviation of Acy damage.

*Hamza, AL-thubaiti & Omar (2019)* demonstrated that Acy afforded significant elevation of MDA with a decline in the antioxidant enzymes activities (SOD, CAT, and GPx) and, accordingly, *Gedik et al. (2017)* observed that Acy administration markedly declined the hepatic glutathione and thus it's administration induced severe hepatic injury and severe hepatocytes damage in structure and appeared clearly in TEM sections and histological sections with appearance of inflammatory cellular infiltration, necrosis and hemorrhage areas in hepatic tissues.

Additionally, Acy-induced excessive production of free radicals due to the breakdown of polyunsaturated fatty acids in the cellular membranes leads to the deterioration of cellular integrity and enhancement of ALT and AST levels. In this study, elevated ALT, AST, and LDH levels were observed in Acy-affected liver cells. This can be attributed to the introduction of ALT, AST, and LDH into the blood circulation, which leads to the destruction of liver cellular membranes due to severe oxidative injury and elevated hepatic enzymes in the blood.

The obtained results of the novel synthesized complex Art/Q/Zn are in accordance with those of a previous study (*Refat et al., 2021*), which reported that pulmonary fibrosis is greatly linked to enhanced oxidative stress and acute respiratory syndromes. Thus, it can be inferred that pulmonary fibrosis is strongly associated with the rapid progression of SARS-CoV-2, which increased the mortality rate during the COVID-19 pandemic. Hence, the modulation of oxidative stress might be an essential concept for curbing SARS-CoV-2 and pulmonary fibrosis. Art/Q/Zn can improve pulmonary fibrosis by upregulating the expression of antioxidant enzyme and elevating the antioxidant activity against oxidative stress markers.

The findings of this study broadly support the antiviral and antioxidant activities of the novel synthesized mixed ligand (Art/Q/Zn), which can normalize high blood pressure and restore hepatic and pulmonary functions and physiological vitality of organs.

## CONCLUSION

This study revealed the chemical structure of a novel mixed ligand Art/Q/Zn. The results of FT-IR analysis showed that Zn (II) can be chelated *via* C=O and C−OH groups present on the Q ligand and C=O present on the Art ligand, thereby forming the Art/Q/Zn complex with the chemical formula $[Zn(Q)(Art)(Cl)(H_2O)_2]\cdot 3H_2O$. The novel complex had high anti–SARS-CoV-2 activity at a low concentration ($IC_{50} = 10.14\,\mu g/ml$) and did not exhibit any cytotoxicity to the host cells ($CC_{50} = 208.5\,\mu g/ml$). It alleviated Acy-induced hepatic and pulmonary toxicity by improving biochemical marker levels, reduced systolic or diastolic blood pressure, and regulated the heart rate after Acy administration.

The Art/Q/Zn complex exhibited antioxidant activity and high anti–SARS-CoV-2 activity. It can also restore organ physiological functions and regulate the interaction of ACE-2 receptor and the $M^{pro}$ target main protease receptor to inhibit the SARS-CoV-2, thereby reducing inflammatory marker levels and its severity during the COVID-19 pandemic.

### Funding

The researchers were funded by the Deanship of Scientific Research, Taif University. The funders had no role in study design, data collection and analysis, decision to publish, or preparation of the manuscript.

### Grant Disclosures

The following grant information was disclosed by the authors:
The Deanship of Scientific Research, Taif University.

### Competing Interests

The authors declare there are no competing interests.

### Author Contributions

- Samy M. El-Megharbel conceived and designed the experiments, performed the experiments, analyzed the data, prepared figures and/or tables, authored or reviewed drafts of the article, and approved the final draft.
- Safa H. Qahl  conceived and designed the experiments, performed the experiments, analyzed the data, prepared figures and/or tables, authored or reviewed drafts of the article, and approved the final draft.
- Bander Albogami conceived and designed the experiments, performed the experiments, analyzed the data, prepared figures and/or tables, authored or reviewed drafts of the article, and approved the final draft.
- Reham Z. Hamza conceived and designed the experiments, performed the experiments, analyzed the data, prepared figures and/or tables, authored or reviewed drafts of the article, and approved the final draft.

### Animal Ethics

The following information was supplied relating to ethical approvals (i.e., approving body and any reference numbers):

The Zagazig University ethical committee approved the study under approval for using Artemisinin, approval number (ZU-IACUC/2/F/61/2022) and the Taif University ethical committee, accredited by the National Committee for Bioethics with No. (HAO-02-T-105).

### Data Availability

The raw data is available at EMBL-EBI: SAMN14814607.
Available at https://www.ebi.ac.uk/ena/browser/view/SRS6581957
Available at https://www.ebi.ac.uk/biosamples/samples/SAMN14814607

### Supplemental Information

Supplemental information for this article can be found online at http://dx.doi.org/10.7717/peerj.15638#supplemental-information.

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
