# Peer review of "Chemical and spectroscopic characterization of (Artemisinin/Querctin/ Zinc) novel mixed ligand complex with assessment of its potent high antiviral activity against SARS-CoV-2 and antioxidant capacity against toxicity induced by acrylamide in male rats"

_PeerJ, doi:10.7717/peerj.15638_

## Round 0.1 · original submission · Major Revisions

Dear authors, thank you for your submission.

Your manuscript presents relevant and interesting work. However, some technical issues need to be overcome before sending out the files to reviewers.

1) language (someone clearly tried to proofread your manuscript and many words were submitted with red font. Nonetheless, it needs extensive language revision. Copyediting is not provided as a standard publication service. Please ensure the language in this submission is clear and unambiguous, grammatically correct and conforms to professional standards of courtesy and expression. Email [email protected] for assistance (we're here to help!));

2) some figures are too small or unclear to be revised / analysed; for example, figure 5. However, it seems like figures 15-21 were constructed by other author(s) and are presented in proper quality and size, and with proper legends. Thus, I suggest to guide your by the quality of these figures and to unify all the figures in quality (minimum dpi 300), font size and type etc and, legends clearly stating what we are looking at and the most important things to observe. Please, do not forget to define acronyms , state scales etc... In terms of figure 6, for example, you also seem to have copy-pasted different chemical notations. This is not correct. Please use freely available docking software or of chemical notation writing to create the formula / structure for Art/Q/Zn. Figure 2 also presents 2 different types of chemical writing. Uniformize this, either present cis-trans for both molecules or the conventional. Also, why only on Figure 6 you present the combined structure (?) of the ligands complex? why not on fig 3 or just together with fig 2? - organisational / structure of the manuscript issue

I cannot emphasise enough that is important to be able to read everything that goes on your figures, thus do not forget to properly label graph axis (including units of measurement) etc. If any figure (eg. figure 1, maybe?) was not created by you, please cite source; if adapted cited it accordingly. Moreover, do NOT forget the controls for all the relevant experiments; this includes in both in vitro and in vivo or ex vivo assays. Any other information refer to authors instructions, but I do believe that this is standard!

Upon improvement of these issues, please upload the revised versions of the files and then, we can proceed to send it for peer-reviewing. Many thanks.

---

## Round 0.2 · Major Revisions

Dear authors thank you for your submission. To be approved for publication your manuscript requires extensive revision and potentially additional experiments. Please, refer to the reviewers' comments for further details.

·

Basic reporting

- The article is written in English and uses clear, unambiguous, technically correct text.
- sufficient introduction and background demonstrate how the work fits into the broader field of knowledge. - Relevant prior literature is appropriately referenced.
- represent an appropriate ‘unit of publication’, and include all results relevant to the hypothesis.

Experimental design

The investigation has been conducted rigorously and to a high technical standard.
The research has been conducted in conformity with the prevailing ethical standards in the field.

Validity of the findings

The conclusions are appropriately stated, and supported by the results.
The relationship is supported by a well-controlled experimental intervention.

Additional comments

Generally, the research is within the scope of the journal.
The manuscript is well written, with proper design and reliable results.
It can be accepted in its current form.

Reviewer 2 ·

Basic reporting

.

Experimental design

.

Validity of the findings

.

Additional comments

1. References 1-3 published in 2020-2021 cannot be cited to describe current COVID-19 epidemiology and available drugs.

Despite the urgent global need and World Health Organization recommendations, an etiotropic therapy of COVID-19 is currently limited and includes 4 drugs: remdesivir, ritonavir-boosted nirmatrelvir (Paxlovid), sotrovimab, and molnupiravir.
Current information is available on the site https://www.covid19treatmentguidelines.nih.gov and is regularly changed.

2. SARS-CoV-2 virions contain envelope, structural proteins and genomic RNA that may be molecular targets for antivirals.
Coronaviruses are spherical enveloped positive single-stranded RNA viruses with a longest single-stranded RNA genome of up to 31 kb in length. The virions consist of structural proteins such as the spike (S), membrane (M), envelope (E), and nucleocapsid (N) proteins. Additionally, there is the hemagglutinin-esterase (HE) protein in some β-coronaviruses. The S, M, and E proteins are embedded in the envelope and the N protein interacts with the viral RNA, forming the nucleocapsid. S protein is necessary for β-coronaviruses to attach to the host cell and enter. The host protease furin cleaves the full-length precursor S glycoprotein into two associated polypeptides: S1 and S2

3. Origin of the virus and cells remain uncertain.
Usually description of viruses include the strain, GenBank accession number of the full-length genome or at least some fragment for phylogenetic analysis, isolation data, natural host and laboratory host(s), passage history.
Additional information about the cells is also required.

4. Paraformaldehyde is polymer and cannot be dissolved in water. Formaldehyde solution with concentrations up to 30% can be obtained at 80 C in alkaline solution.

5. Total number of illustrations (24 figures and 6 tables) exceeds limits for research papers and is more convenient for book chapters. Some of them might be shown in Supplementary data.


Minor comments
with numerous spelling mistakes and unreproducible results because of inexact methods are marked in the attached file.

Annotated reviews are not available for download in order to protect the identity of reviewers who chose to remain anonymous.

Reviewer 3 ·

Basic reporting

It is a good work and the study design is OK. The major issues are with the synthesis of the compound. The complexation is not confirmed with the present analysis (2.3. Synthesis of the mixed ligand zinc complex).
It is very important to see the 1H NMR of both ligand and metal complex. Please do and keep in the text.
Mass spectra for both ligand and metal complex is also required. Please include in the text.
Why authors not tried for single crystal XRD? Why powder XRD, SEM, EDX, TEM. Is it confirm the synthesis.
After synthesis --- then other applications -- we may conclude.
If synthesis is not confirmed---it will be hypothetical.
Some old references are cited in the introduction. Add very recent references (DOI: 10.1111/CBDD.14035; DOI: 10.1016/j.jscs.2021.101367; DOI: 10.1016/j.molstruc.2021.132039; DOI: 10.1016/j.jksus.2022.102086; DOI: 10.1016/j.molstruc.2022.134676; DOI: 10.3389/fviro.2022.1077155; DOI: 10.1097/JS9.0000000000000071).
Further, the introduction should be modified. Delete---line-47 to 66. Please add a literature review as per your study design.
As per toxicity data, authors are silent about its administration.

Experimental design

Please check the above-2.3. Synthesis of the mixed ligand zinc complex

Validity of the findings

Synthesis is not confirmed.

Additional comments

Please revise accordingly.

---

## Round 0.3 · Major Revisions

Dear authors,

I regret to inform you that certain changes have been brought to my attention. While these changes do not affect the scientific question addressed in your manuscript, they do not meet the minimal standards of quality for a scientific publication. To ensure the suitability of your work for publication, the following actions are needed:

Fig 10: The original source must be properly cited. Please ensure that this article is attributed correctly, as it is available under the Creative Commons CC-BY-NC-ND license, allowing non-commercial use without adaptation or alteration.

Figure 15: Do not remove B2. Utilize Figure 15-2 with B2. Ensure that the scales used in A/C, B1/D, and B2 are consistent. Figures must be replaced with images measuring between 900 and 3000 pixels on all sides, saved as PNG or vector PDF format without excess white space.

Figure 18: Keep either the original or 18-2 (A, B1, B2, C1, C2). If using 18-2, correct the scales on all parts, considering the zoomed-in aspect of part C.

Figure 22: Revise the legend to ensure accuracy and clarity.

Figure 23: Include all three parts of the figure in a single file. Revise the legend and the text in the manuscript related to Figure 23, addressing any grammatical errors and improving its structure and presentation. Correct the citation inconsistency (Fig 23 vs. Fig 24, A1 vs. B1).

Figure 24: Ensure that all five parts of Figure 24 are included, improving the quality of the figure if necessary. Correct the citation inconsistency (fig. 24 A3 vs. Fig A2).

It is crucial to correct any errors in the figures, such as the inconsistent scales in histology pictures. Replacement figures must adhere to the specified dimensions and format guidelines. Please, also note that the office does not provide language services, therefore CAREFULLY proofread and address any ill-prepared language in your manuscript. Carefully revise your statistics and their presentation.

I will personally review the final manuscript you present (pdf), therefore, make sure this meets all of our requirements (namely in terms of figures quality) and that all figures and tables and any schematics are presented as intended to.

Please address these issues promptly to bring your manuscript up to the required standards for publication. Your attention to these matters will greatly contribute to the overall quality and credibility of your research.
Many thanks in advance.

---

## Round 0.4 · Minor Revisions

Dear authors,

I / We extend our gratitude for your diligent efforts and hard work in preparing the manuscript. However, at this stage, I regret to inform you that I am unable to provide approval for publication of the materials. While you have provided an English language certificate from a professional proof-editing service, the quality of the language still requires improvement. I recommend that you revisit the service provider and request a revision. It is essential that they demonstrate proficiency in academic English for effective communication.

There are several instances of typos, misplaced spaces, and formatting issues that need to be addressed. The approach you undertook for improving Figure 24's quality and resolution should also be applied to earlier images to maintain consistency.

I have attached a PDF containing detailed notes outlining these concerns. It's important to note that my review hasn't covered the entire manuscript, but even in the excerpts I've examined, there were several problematic aspects. I kindly urge you to be thorough, clear, and comprehensive in your revisions to ensure the manuscript achieves the highest possible readability. Many thanks in advance.

(PS. I found typos in the references but, I did not check for the authors' names' spelling, so please, be really careful in your editing/ proofreading)

---

## Round 0.5 · Minor Revisions

Dear authors,

I have reviewed your manuscript and wanted to provide some feedback namely regarding language and formatting issues that still appear to be present. While the overall quality of the figures seems improved, it's essential to ensure they meet the required standards. Additionally, I noticed what may be typos in figures 4 and 5.

I understand that the editorial team does not offer language services, and there may have been an attached certificate, the authenticity of which I cannot confirm. Nevertheless, addressing these small issues can significantly enhance the readability and professionalism of your manuscript. Thus, I suggest you go back to the company you hired for the language services and demand a full, proper, re-do! Remind them that academic English and literature or colloquial English are slightly different.

I acknowledge and remind you that repeatedly sending the manuscript back for the same fixes can be tedious, especially considering that the editorial team is not specifically tasked with such revisions.

Please consider reviewing the manuscript once more, paying particular attention to things like proper comma and bracket usage, article placement, and ensuring the accuracy and consistency of all figures.

---

## Round 0.6 · Minor Revisions

Dear authors, regrettably, we are still encountering challenges in ensuring that you grasp the essential publication requirements, particularly regarding typos and the quality of figures' content. I strongly encourage you to comprehensively address the following issues:
There are many problems with your figures that must be addressed completely before they are ready for production. Please address all the issues below before resubmitting:

1: Please remove all figure numbers, titles, and legends from the figure files. This information should not be contained in the figure file.
- Figure 4: Remove "Timeline of the experimental design" and "Experimental timeline for (Acy) and novel complex (Art/Q/Zn) treatment
- Figure 23: Remove "3D Binding interaction (Art/Q/Zn)" from Part A, "Simulation of novel complex of (Art/Q/Zn) with ACE2 receptor", "3D simulation of Art/Q/Zn with ACE2" and "Lipophilicity of Art/Q/Zn with ACE2" from Part B, and "Simulation of novel complex of (Art/Q/Zn) with MPro receptor", "3D simulation of Art/Q/Zn with Mpro" and "Lipophilicity of Art/Q/Zn with Mpro"from Part C. This text should only appear in the legend in the system if it is necessary to include with the figures: "Systolic and diastolic blood pressure in Acy and /or Art/Q/Zn treated group 3D binding interaction of novel complex (Art/Q/Zn) with ACE2 and MPro receptors."

2: There are typos in your figures.
- Figure 4: "Measurement" is misspelled two times.
- Figure 5: "administration" is misspelled. "immunoassay" must be one word.
- Figure 8: "wavelength" must be one word

3: Some of your figures cannot be viewed properly as they are pixelated and difficult to read.

- Figure 9: Please supply a high quality replacement.
- Figure 10: Please supply a high quality replacement.
- Figure 24: Please supply a high quality replacement.

4: Every figure with multiple parts must have alphabetical (e.g. A, B, C) labels, not numerical labels, on each part and all parts of each single figure should be submitted together in one file.

- The 6 parts of Figure 11 must be labeled A-F to correspond with the legend: "SEM of (A, B) Art, (C, D) Q and (E, F) Art/Q/Zn"
- The 6 parts of Figure 13 must be labeled A-F to correspond with the legend: "TEM of (A, B) Art, (C, D) Q and (E, F) Art/Q/Zn"
- The 5 parts of Figure 15 must be labeled A-E to correspond with the legend: "(A) Control group showing normal hepatic structure with normal hepatocytes ... (B) Acy treated group showing hepatotoxicity by hypertrophy ... (C) A cross section of liver tissues after Acy administration ... (D) A cross section of liver tissues after (Art/Q/Zn) administration showing normal hepatic structure .... (E) A cross section of liver tissues after Acy administration followed by the novel complex"
- The 5 parts of Figure 16 must be labeled A-E to correspond with the legend: "An electron micrograph of the hepatic tissues of control group (A) which showed a normal hepatic structure .... (B) Acy treated group showed large red blood cells... (C) Acy treated group showed high fatty change (White arrow) with some destructed mitochondria .... (D) Art/Q/Zn treated group showing normal hepatic structures with enlarged ... (E) Acy plus novel mixed ligand complex (Art/Q/Zn) ..."
- The 5 parts of Figure 18 must be labeled A-E to correspond with the legend: "(A) Control group showing the normal perivascular cells.... (B) Acy treated group showing pulmonary toxicity .... (C) Sections of rat lung treated with Acy administration (D) A cross section of rat lung after novel complex (Art/Q/Zn) administration... (E) A cross section of rat lung after administration of Acy ...”
- The 8 parts of Figure 21 must be labeled A-H to correspond with the legend: "A: Lung tissues of Acy group. B: Lung tissues of ART/Q/Zn group. C, D, E: Hepatic tissues of Acy treated group. F, G, H: Hepatic tissues of ART/Q/Zn treated group."
- The 10 parts of Figure 24 must be labeled A-J

Please provide replacement Figures 4, 5, 8, 9, 10, 11, 13, 15, 16, 18, 21, 23, 24 measuring minimum 900 pixels and maximum 3000 pixels on all sides, saved as PNG, EPS, or PDF (vector images only) file format without excess white space around the images. The file name should be formatted as "Figure1.png".

---

## Round 0.7 · accepted · Accept

Dear authors, thank you for all your hardwork and efforts. It was a hard labour but i am happy to let you know that your manuscript has now been approved for publication in PeerJ.